# High-content microscopy reveals a morphological signature of bortezomib resistance

Megan E Kelley[1], Adi Y Berman[1], David R Stirling[2], Beth A Cimini[2], Yu Han[2], Shantanu Singh[2†], Anne E Carpenter[2*†], Tarun M Kapoor[1*†], Gregory P Way[2,3*†]

[1]Laboratory of Chemistry and Cell Biology, The Rockefeller University, New York City, United States; [2]Imaging Platform, Broad Institute, Cambridge, United States; [3]Department of Biomedical Informatics, University of Colorado Anschutz Medical Campus, Aurora, United States

*For correspondence:
anne@broadinstitute.org (AEC);
kapoor@rockefeller.edu (TMK);
Gregory.way@cuanschutz.edu
(GPW)

†These authors contributed
equally to this work

Competing interest: The authors
declare that no competing
interests exist.

Reviewing Editor: Caigang Liu,
Shengjing Hospital of China
Medical University, China

**Abstract** Drug resistance is a challenge in anticancer therapy. In many cases, cancers can be resistant to the drug prior to exposure, that is, possess intrinsic drug resistance. However, we lack target-independent methods to anticipate resistance in cancer cell lines or characterize intrinsic drug resistance without a priori knowledge of its cause. We hypothesized that cell morphology could provide an unbiased readout of drug resistance. To test this hypothesis, we used HCT116 cells, a mismatch repair-deficient cancer cell line, to isolate clones that were resistant or sensitive to bortezomib, a well-characterized proteasome inhibitor and anticancer drug to which many cancer cells possess intrinsic resistance. We then expanded these clones and measured high-dimensional single-cell morphology profiles using Cell Painting, a high-content microscopy assay. Our imaging-and computation-based profiling pipeline identified morphological features that differed between resistant and sensitive cells. We used these features to generate a morphological signature of bortezomib resistance. We then employed this morphological signature to analyze a set of HCT116 clones (five resistant and five sensitive) that had not been included in the signature training dataset, and correctly predicted sensitivity to bortezomib in seven cases, in the absence of drug treatment. This signature predicted bortezomib resistance better than resistance to other drugs targeting the ubiquitin-proteasome system, indicating specificity for mechanisms of resistance to bortezomib. Our results establish a proof-of-concept framework for the unbiased analysis of drug resistance using high-content microscopy of cancer cells, in the absence of drug treatment.

## Editor's evaluation

In this work, the authors have profiled the morphological signatures of the HCT116 cell line and correlated them with bortezomib treatment response, which could provide novel insight into the research of resistance in multiple myeloma from the perspective of morphology. The findings are supported by solid evidence and sufficient experimental validation.

## Introduction

Targeted cancer therapies often fail due to drug resistance, which makes determining the drug sensitivity of populations of cancer cells requisite for timely and effective treatment (*Garraway and Jänne, 2012*; *Pisa and Kapoor, 2020*; *Vasan et al., 2019*). Resistance is complex and can be categorized as acquired, manifesting in the context of prolonged treatment, or intrinsic, pre-existing in the cancer cell population (*Gottesman et al., 2016*). Most studies of resistance focus on the drug's known target(s)

or expression levels of drug efflux pumps (*Garraway and Jänne, 2012*; *Gottesman et al., 2016*). However, we currently lack unbiased methods of identifying drug resistance in cells especially prior to treatment.

Bortezomib is an anticancer drug commonly used to treat multiple myeloma and nearly half of multiple myeloma patients show no initial response to bortezomib therapy, indicating intrinsic resistance (*Chen et al., 2011*; *Gonzalez-Santamarta et al., 2020*; *Mitsiades et al., 2004*; *Hideshima et al., 2001*; *Vincenz et al., 2013*). Bortezomib resistance can be attributed to targeted mechanisms such as mutations in the bortezomib-binding pocket of the proteasome subunit (PSMB5) and overexpression of proteasome subunits (*Barrio et al., 2019*; *Franke et al., 2012*; *van de Ven et al., 2008*; *Wacker et al., 2012*) as well as non-specific mechanisms, such as upregulation of prosurvival or anti-apoptotic pathways and enhanced cell adhesion to the extracellular matrix (*Gonzalez-Santamarta et al., 2020*; *Hideshima et al., 2007*). A priori knowledge of tumor cells' susceptibility to candidate therapeutics could aid in identifying effective treatment options, resulting in fewer relapses and failed treatments due to resistance. However, current methods for evaluating drug resistance depend on viability assays and sequencing, which may be limited in its usefulness without knowledge of the full spectrum of resistance-conferring genomic alterations (*Wheler et al., 2014*) or knowing specific mutations or indels in the target that suppress drug activity (*Kapoor and Miller, 2017*). Methods for determining tumor cell susceptibility prior to therapy are desirable.

A growing literature suggests that specific genomic alterations, treatment response, and prognosis can be predicted from conventional hematoxylin and eosin tissue slides using machine learning (*Cifci et al., 2022*; *Lee and Jang, 2022*), indicating that image data holds promise for predicting drug resistance. High-content microscopy, which uses cell-based automated microscopy to capture information-rich images, has successfully categorized small molecule inhibitors by their mechanisms and targeted pathways (*Ljosa et al., 2013*; *Perlman et al., 2004*) and shown a relationship between morphological profiles and genetic perturbations (*Rohban et al., 2017*), including specific mutations associated with lung cancer (*Caicedo et al., 2022*). This profiling method often uses high-throughput microscopy, generating a large amount of image data from which thousands of quantitative, single-cell morphological features can be extracted to characterize signals that could not be discovered using low-throughput methods and would otherwise be impossible to study by eye. However, high-content microscopy has not been used to examine the features of resistance in the absence of drug treatment.

Here, we used Cell Painting (*Bray et al., 2016*), a multiplex, fluorescence microscopy assay that labels eight cellular components using six stains imaged in five channels, as an unbiased method to characterize the morphological differences between bortezomib-resistant and -sensitive cancer cells. We applied a reproducible imaging- and computation-based profiling pipeline to process the images and identify a high-dimensional cell morphology signature to predict bortezomib resistance that we evaluated using machine learning best practices. This morphological signature correctly predicted the bortezomib resistance of seven out of ten clones not included in the signature training dataset. Overall, our results establish a proof-of-concept framework for identifying unbiased signatures of drug resistance using high-content microscopy. The ability to identify drug-resistant cells based on morphological features provides a valuable method for characterizing resistance in the absence of drug treatment.

## Results
### Isolating and capturing Cell Painting profiles for HCT116-based bortezomib-resistant clones

We first isolated and characterized drug-resistant cells (*Figure 1A*). To isolate drug-resistant clones, we used an approach we have described previously (*Kasap et al., 2014*; *Wacker et al., 2012*) and the HCT116 cell line. These cancer cells express multidrug resistance pumps at low levels and are mismatch repair deficient, providing a genetically heterogeneous polyclonal population of cells (*Papadopoulos et al., 1994*; *Teraishi et al., 2005*; *Umar et al., 1994*) allowing for isolation of drug-resistant clones in ~2–3 weeks. We hypothesized that a rapid selection of resistance could favor the isolation of clones with intrinsic resistance. To determine the appropriate drug concentrations to use in order to isolate drug-resistant clones, we performed proliferation assays on HCT116 parental cells with

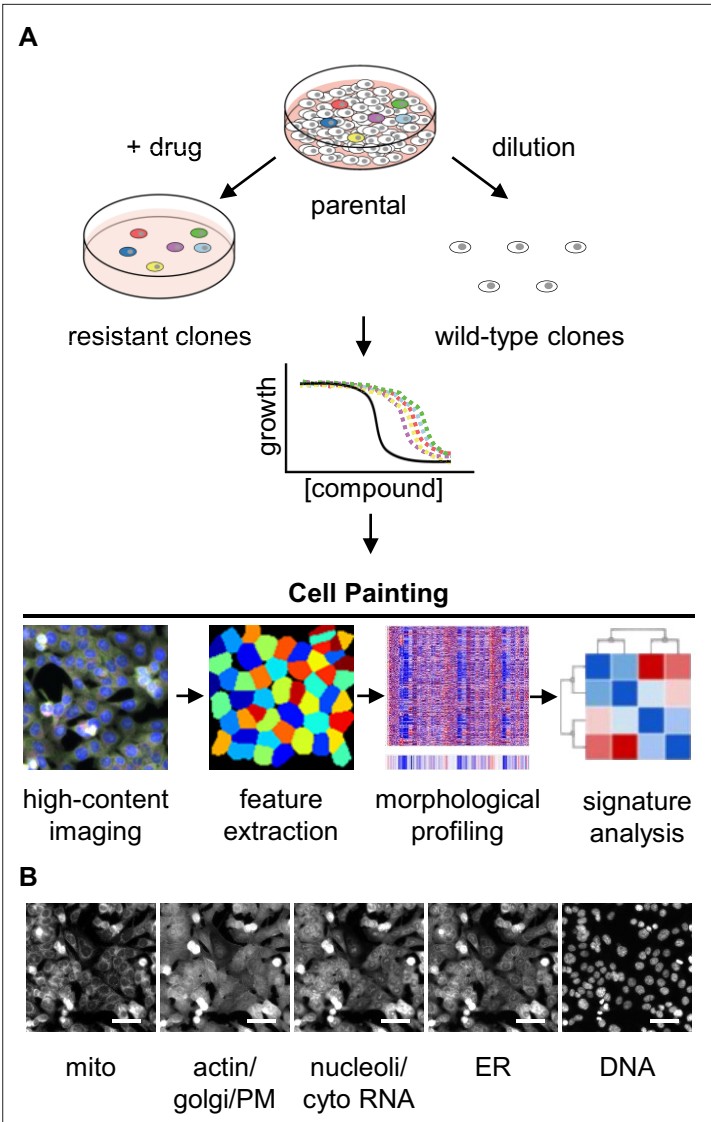

**Figure 1.** Experimental design for using Cell Painting to examine morphological profiles of drug-resistant cells. (**A**) Graphic of the experimental workflow: we isolated drug-resistant clones by treating parental HCT116 cells with the desired drug and then expanded them for experiments. We isolated drug-sensitive clones by diluting HCT116 cells and then expanded them for experiments. We then performed proliferation assays on select clones to evaluate them for multidrug resistance. Next, we performed Cell Painting on both drug-resistant and -sensitive clones, using multiplexed high-throughput fluorescence microscopy of fixed cells followed by feature extraction and morphological profiling to search for features that contribute to a signature of drug resistance. (**B**) One representative field of view of cells labeled with six fluorescent dyes and captured in five channels used for morphological profiling with Cell Painting. Scale bars, 50 μm.

The online version of this article includes the following source data and figure supplement(s) for figure 1:

**Source data 1.** LD50s for HCT116 cells.

**Source data 2.** Cell line descriptions.

**Figure supplement 1.** Proliferation assays of HCT116 parental cells in drugs used to isolate resistant clones.

**Figure supplement 2.** Bortezomib-resistant clones do not display strong features of multidrug resistance.

our drugs of interest: bortezomib, ixazomib, or CB-5083 (*Figure 1—figure supplement 1A–C* and *Figure 1—source data 1*). We also isolated bortezomib-sensitive (wild-type; WT) clones by dilution of the HCT116 parental cell line and acquired two published bortezomib-resistant clones (BZ clones A and E) both with mutations in PSMB5 identified by RNA sequencing performed in previous work

(*Figure 1—source data 2*; *Wacker et al., 2012*). We characterized the bortezomib-resistant clones and found that the median lethal doses (LD50s) for bortezomib were ~2.8- to ~9-fold that of HCT116 parental cells (*Figure 1—figure supplement 2B*). In contrast, bortezomib-sensitive clones had LD50s for bortezomib that ranged from ~0.7- to~1.2-fold that of HCT116 parental cells (*Figure 1—figure supplement 2A*). Together these methods provided a total of twelve bortezomib-resistant, five ixazomib-resistant, five CB-5083-resistant, and twelve bortezomib-sensitive clones as well as HCT116 parental cells for our experiments.

To screen for multidrug resistance, which might convolute a specific signature of bortezomib resistance, we measured proliferation of the bortezomib-resistant and -sensitive clones in the presence of two drugs with different mechanisms of action: taxol (a microtubule poison) and mitoxantrone (a topoisomerase inhibitor) (*Liu, 1989*). Bortezomib-resistant and -sensitive clones treated with taxol had LD50s ranging from ~0.6- to~1.9-fold that of HCT116 parental cells (*Figure 1—figure supplement 2C and D*). Treating cells with mitoxantrone, we found that the bortezomib-sensitive clones (*Figure 1—figure supplement 2E*) and most of the bortezomib-resistant clones had similar LD50s (*Figure 1—figure supplement 2F*). There was one exception (BZ06) that had an LD50 nearly 14-fold higher than that of HCT116 parental cells.

We next applied the Cell Painting assay to all these drug-sensitive and -resistant clones. Cell Painting captures signal in five imaging channels from six fluorescent dyes that stain cells for eight cellular components including mitochondria, actin, Golgi, plasma membrane, cytoplasmic RNA, nucleoli, endoplasmic reticulum, and DNA (*Figure 1B*; *Bray et al., 2016*). With these images, we used CellProfiler (*Stirling et al., 2021*) to extract single-cell morphological features from individual cells. The signal from each of the five channels was analyzed in the nucleus, cytoplasm, and total cell and characterized based on features (object parameters) such as signal intensity, shape of the object, texture of the staining pattern, etc. yielding a total of ~3500 features. These cellular features were combined and analyzed on a per well basis (well profiles) and then compared across cells and experimental conditions to determine whether morphological features of drug resistance could be reliably detected in the absence of drug treatment.

## A subset of morphological features contribute to the signature of bortezomib resistance

To examine whether there were any clear qualitative morphological differences between bortezomib-resistant and -sensitive cells we chose HCT116 parental cells, bortezomib-sensitive clones WT01-WT05, and bortezomib-resistant clones A, E, and BZ01-BZ05 for our initial studies. We treated cells with 0.1% DMSO (to allow for comparison with future experiments using drug-treated cells) and performed Cell Painting, staining fixed cells and imaging as per the published protocol (*Bray et al., 2016*). We observed cellular heterogeneity within each clone as well as between clones with similar bortezomib sensitivities (*Figure 2A* and *Figure 2—figure supplement 1*). This heterogeneity obscured any potential morphological differences between clones and prevented us from qualitatively distinguishing bortezomib-resistant from -sensitive clones by eye, supporting the need for high-content quantitative analysis.

We then pre-processed profiles to remove low-variance and highly correlated features, and population-averaged single cell measurements at the well level to generate well profiles (see Materials and methods). The morphological profiles of bortezomib-resistant and bortezomib-sensitive cells did not cleanly distinguish clones based on bortezomib resistance (*Figure 2—figure supplement 2A*). We saw a similar failure to distinguish clones based on bortezomib sensitivity after a short, 4 hr treatment with 7 nM bortezomib (*Figure 2—figure supplement 2B*), suggesting that if there is a morphological difference between bortezomib-resistant and -sensitive cells, further feature refinement would be needed for its identification.

Each observed morphological measurement results from a combination of both technical and biological variables. It is therefore important to control and test for technical variables as these can confound subtle biologically relevant signatures. Using bortezomib-sensitive clones WT01-05 and bortezomib-resistant clones BZ01-05 to quantify and reduce the impact of technical variables, we fit a linear model to each morphological feature adjusting for technical variables (experimental run/batch, incubation time, cell count/density, clone ID) and biological variables (resistance status) (see Materials and methods). We then discarded morphological features with variances that correlated

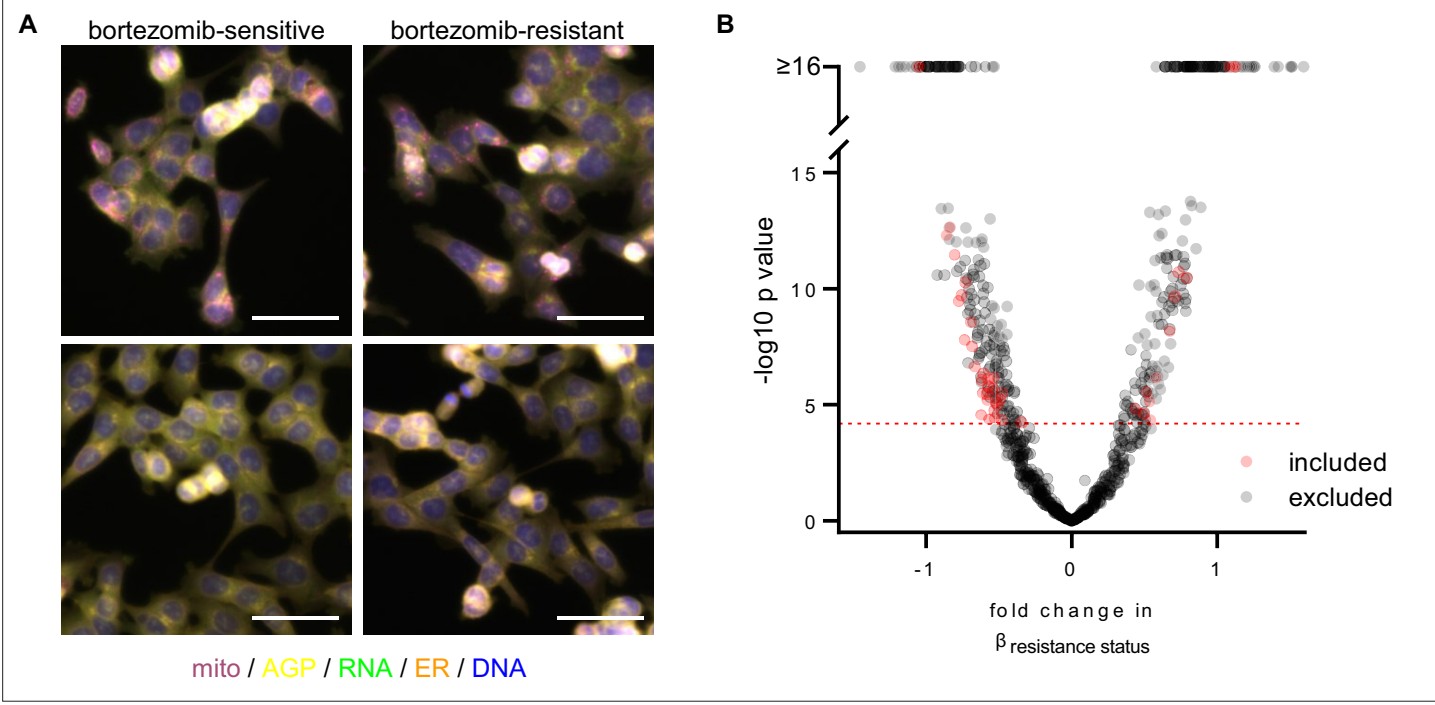

**Figure 2.** A subset of morphological features contributes to the signature of bortezomib resistance. (**A**) Representative fixed fluorescence microscopy images of two bortezomib-sensitive (WT02 and WT03) and two bortezomib-resistant (BZ02 and BZ03) clones stained and imaged as per the Cell Painting protocol. Channels are labeled as mito (mitochondria; magenta), AGP (actin, golgi, plasma membrane; yellow), RNA (ribonucleic acid; green), ER (endoplasmic reticulum; orange), and DNA (deoxyribonucleic acid; blue). See *Figure 2—figure supplement 1* for single-channel images. Scale bars, 50 μm. (**B**) Volcano plot of the variability of morphological features (β) by resistance status. Y-axis -$\log_{10}$p values are from Tukey's Honestly Significant Difference test score (see Materials and methods). Red circles are features included in the final signature of resistance and gray circles are features excluded from the final signature. Features above the red dashed line (-$\log_{10}$[0.05/number of unique features]) were considered significantly varying and those that had not been excluded as technical variables (*Figure 2—figure supplement 3*) were included in the signature of bortezomib resistance. n = 6 independent experiments (biological replicates).

The online version of this article includes the following source data and figure supplement(s) for figure 2:

**Source data 1.** A subset of CellProfiler features contribute to the signature of bortezomib resistance.

**Figure supplement 1.** Individual channels of bortezomib-sensitive and -resistant clones imaged by Cell Painting.

**Figure supplement 2.** Similarity clustering is insufficient to distinguish bortezomib-resistant from -sensitive clones.

**Figure supplement 3.** Technical variables are controlled for and excluded from analyses.

**Figure supplement 4.** Features contributing to the Bortezomib Signature do not universally correlate.

**Figure supplement 5.** Bortezomib Signature visualized by CellProfiler features.

**Figure supplement 6.** Well location does not strongly correlate with Bortezomib Signature.

**Figure supplement 7.** Bortezomib Signature identifies resistant clones and not technical variables.

with experimental run (batch), incubation time (4 or 13 hours with 0.1% DMSO), cell density, or those features that varied between two or more pairs of bortezomib-sensitive clones (clone ID) (*Figure 2—figure supplement 3A-E*). Of the remaining morphological features, we only considered those that varied based on the resistance status of a clone (*Figure 2B*). This resulted in 45 morphological features that significantly contributed to a clone's bortezomib resistance (*Figure 2—source data 1*). We used these 45 features to compute a rank-based resistance score or 'Bortezomib Signature' for each well profile based on the direction-sensitive method called *singscore* (*Foroutan et al., 2018*). *Singscore* ranks these 45 resistance-related features on a per sample basis and calculates a normalized score between –1 and 1, with higher values expected for bortezomib-resistant clones and lower values expected for bortezomib-sensitive clones. With the exception of some texture-based features, the Bortezomib Signature features were largely independent, displaying low pairwise correlation (*Figure 2—figure supplement 4*). We then examined the grouping of features across compartments

and channels and found radial distribution features were higher in resistant cells (*Figure 2—figure supplement 5*). Anticipating well location as a possible technical artifact, we plated our cells in a repeating serpentine pattern, ensuring that each clone would be imaged in multiple locations on a plate (*Figure 2—figure supplement 6A*). We found that the pattern of Bortezomib Signatures did not correspond to well position relative to the plate (*Figure 2—figure supplement 6B*), indicating that the well position for each clone was not strongly contributing to its Bortezomib Signature. In addition, we found that the Bortezomib Signature correlated with the resistance status of clones and not technical variables (*Figure 2—figure supplement 7*). These data suggest that our analysis pipeline and signature building process minimized technical artifacts.

## Cell morphology predicts the bortezomib response of multiple clones across datasets

To evaluate the performance of the Bortezomib Signature, we used machine learning best practices, separating our data into training, validation, test, and holdout datasets (*Figure 3—source data 1*; see Materials and methods). The data initially used to create the Bortezomib Signature, which included well-based morphological profiles from clones WT01-05 and BZ01-05, was designated as the training dataset. The validation dataset was composed of well profiles from clones WT01-05 and BZ01-05 that were not used to generate the Bortezomib Signature but were collected on the same plates as the well profiles used for the training dataset. The test dataset was composed of well profiles from HCT116 parental cells and bortezomib-resistant clones A and E; these cells were not included in the training dataset, but their well profiles were collected on the same plates as those used for the training dataset. The holdout dataset was a separate plate and contained HCT116 parental cells, bortezomib-sensitive clones WT01-05, and bortezomib-resistant clones A, E, and BZ01-05. These datasets allowed us to test generalizability across clones and plates for the Bortezomib Signature.

We next examined whether the Bortezomib Signature was able to predict the bortezomib resistance of a clone based on morphological profiling data (*Figure 3A–E* and *Figure 3—figure supplement 1A and B*). We called the clone bortezomib-resistant if the median Bortezomib Signature of all replicate well profiles was greater than zero and bortezomib-sensitive if the median Bortezomib Signature was less than zero. In the training dataset, the Bortezomib Signature correctly predicted the bortezomib resistance of all ten clones, with median Bortezomib Signatures for eight out of ten clones beyond the 95% confidence interval for the randomly permuted data (*Figure 3A*). The accuracy of the Bortezomib Signature was 88% while the average precision was 81% for the training dataset (*Figure 3—figure supplement 1A and B*) (see Materials and methods). The signature performed similarly well in the validation dataset (*Figure 3B*), with an accuracy of 92% and an average precision of 89% (*Figure 3—figure supplement 1A and B*). In the test dataset the Bortezomib Signature correctly predicted the bortezomib resistance of all clones, although only HCT116 parental cells had a median Bortezomib Signature outside the 95% confidence interval for the randomly permuted data (*Figure 3C*). The test dataset had an accuracy of 80% and an average precision of 68% (*Figure 3—figure supplement 1A and B*). Similarly, in the holdout dataset the Bortezomib Signature had an accuracy of 78% and an average precision of 69% (*Figure 3—figure supplement 1A and B*), and correctly predicted the bortezomib resistance of twelve out of thirteen clones, with WT01 misclassified as bortezomib-resistant (*Figure 3D*). In the holdout dataset, four of the twelve correctly characterized clones had median Bortezomib Signatures outside the 95% confidence interval for the randomly permuted data. The Bortezomib Signature performed better than random chance in all testing conditions, as demonstrated by comparison with the mean accuracy and average precision for the randomly shuffled data (*Figure 3—figure supplement 1A and B*), and as reflected in receiver operating characteristic (ROC) curves, which describe the classification trade-off between true positive and false positive rates in predicting bortezomib resistance (*Figure 3E*). We then compared our linear-modeling approach to feature selection against other feature spaces and found that the Bortezomib Signature clusters same-type clones (bortezomib-resistant vs. bortezomib-sensitive) with higher enrichment compared to the full feature space, standard feature selection (see Materials and methods), or a random selection of 45 features (*Figure 3—figure supplement 2*). These data are consistent with the Bortezomib Signature being able to distinguish bortezomib-resistant from -sensitive clones better than random chance across datasets.

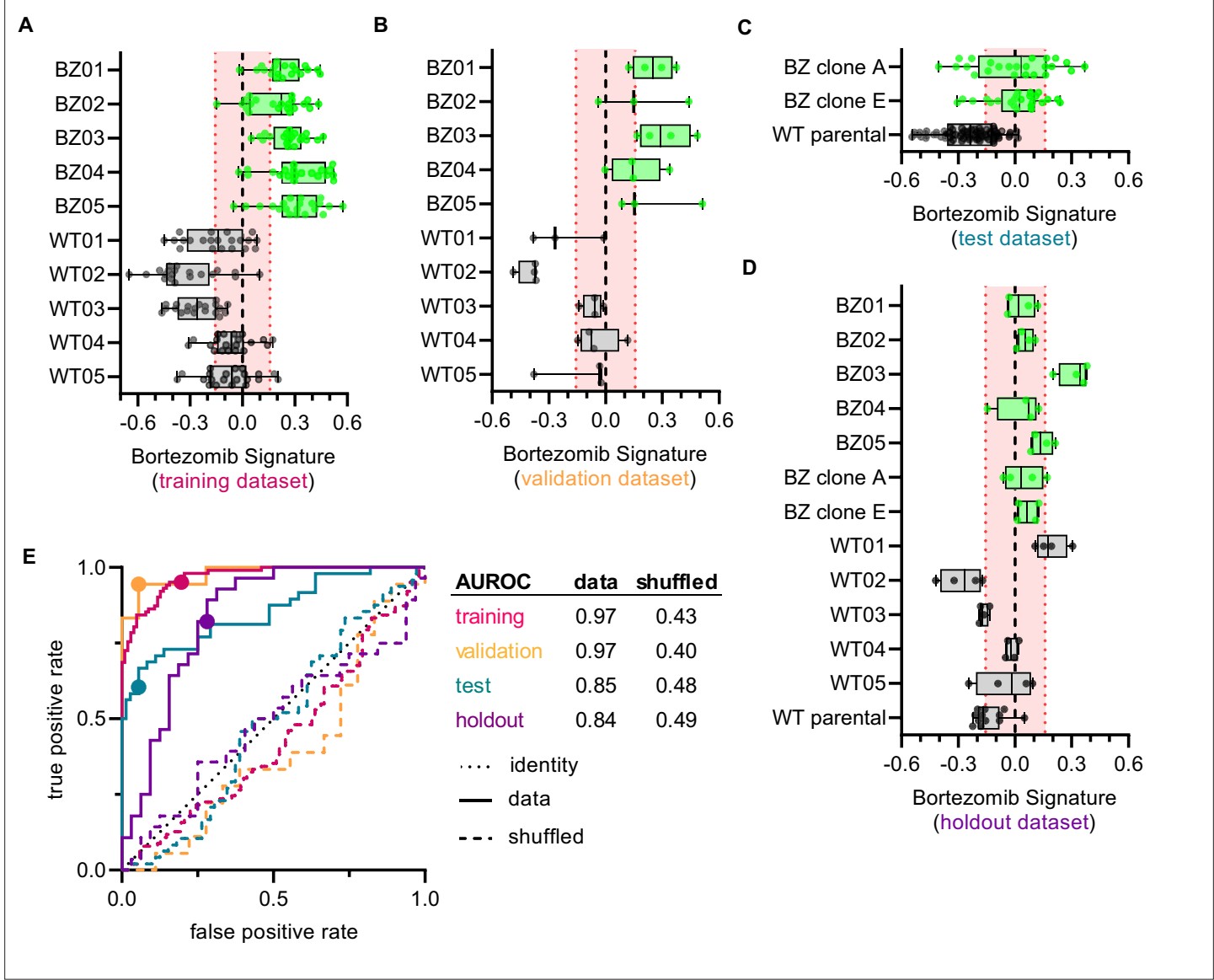

**Figure 3.** Cell morphology predicts the bortezomib sensitivity of clones across datasets. Box plots of Bortezomib Signatures for clones in the (**A**) training, (**B**) validation, (**C**) test, and (**D**) holdout datasets. Plots show values for individual well profiles (points), range (error bars), 25th and 75th percentiles (box boundaries), and median. Dashed vertical black line is Bortezomib Signature = 0, dashed vertical red lines are the 95% confidence interval for Bortezomib Signatures of 1000 random permutations of the data. (**E**) ROC curves for the performance of the Bortezomib Signature on the indicated dataset (solid line) or its shuffled counterpart (dashed line). Datasets are designated by color: training (magenta), validation (orange), test (teal), and holdout (purple). Colored points are the corresponding false positive and true positive rates at the absolute minimum thresholds for each respective dataset. Black dotted line is the identity line where false positive rate = true positive rate. AUROC values reported for data and shuffled data. See *Figure 3—source data 1* for breakdown of profiles and experiments per dataset.

The online version of this article includes the following source data and figure supplement(s) for figure 3:

**Source data 1.** Datasets for Bortezomib Signature generation and evaluation.

**Figure supplement 1.** Accuracy and average precision of the Bortezomib Signature.

**Figure supplement 2.** Benchmarking linear-modeling feature selection to separate clones by bortezomib resistance.

## Bortezomib Signature has moderate specificity for bortezomib over other ubiquitin-proteasome system inhibitors

To examine whether the Bortezomib Signature was specific to the drug bortezomib or was a general signature of UPS-targeting drug resistance we performed Cell Painting on HCT116 clones that

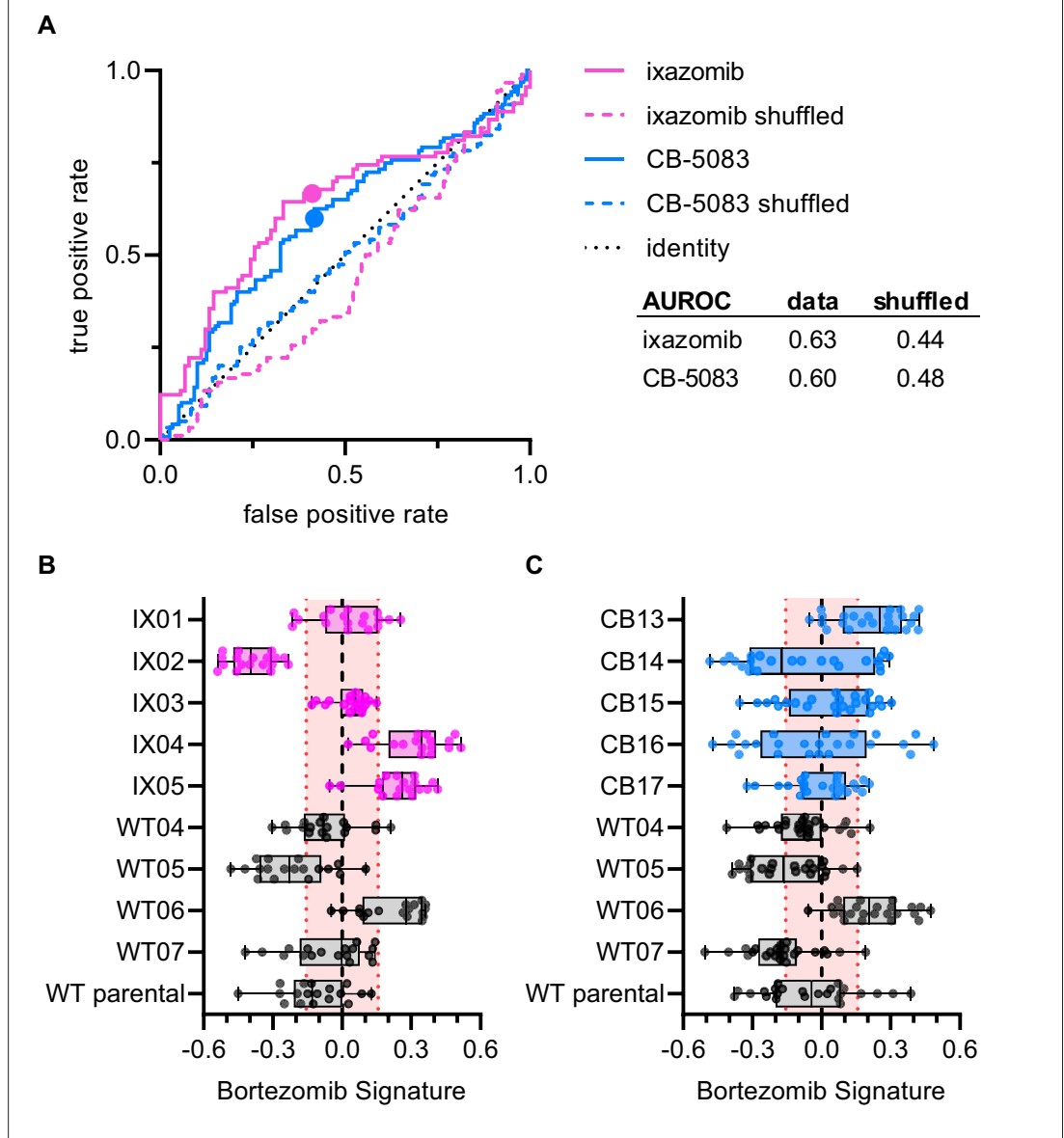

**Figure 4.** Bortezomib Signature has limited ability to characterize clones resistant to other UPS-targeting drugs. (**A**) ROC curves for ixazomib-resistant (magenta) and CB-5083-resistant (blue) experimental data. Colored solid lines are the actual data while colored dashed lines are the shuffled data for each set of clones. Colored points are the corresponding false positive and true positive rates at the absolute minimum thresholds for each respective cell type. Black dotted line is the identity line where false positive rate = true positive rate. AUROC reported for the data and shuffled data. Box plots of Bortezomib Signatures for (**B**) ixazomib-resistant and bortezomib-sensitive clones (n = 18 profiles, 3 independent experiments) and (**C**) CB-5083-resistant and bortezomib-sensitive clones (n = 24 profiles, 4 independent experiments). Plots show values for individual well profiles (points), range (error bars), 25th and 75th percentiles (box boundaries), and median. Dashed vertical black line is Bortezomib Signature = 0, dashed vertical red lines are the 95% confidence interval for Bortezomib Signatures of 1000 random permutations of the data.

were resistant to either ixazomib (another proteasome inhibitor that targets the PSMB5 subunit) or CB-5083 (a p97 inhibitor that acts upstream of the proteasome). If the Bortezomib Signature were a general signature of UPS-targeting drug resistance, we would expect it to perform equally well at characterizing the drug sensitivity of bortezomib-, ixazomib-, and CB-5083-resistant clones. The Bortezomib Signature performed better than chance at identifying ixazomib-resistant and CB-5083-resistant clones (*Figure 4A*), correctly identifying four of five ixazomib-resistant clones (*Figure 4B*) and three of five CB-5083-resistant clones (*Figure 4C*). However, only two of the four correctly identified ixazomib-resistant clones and one of the three CB-5083-resistant clones had median Bortezomib Signatures outside the 95% confidence interval of the randomly permuted data. The area under the

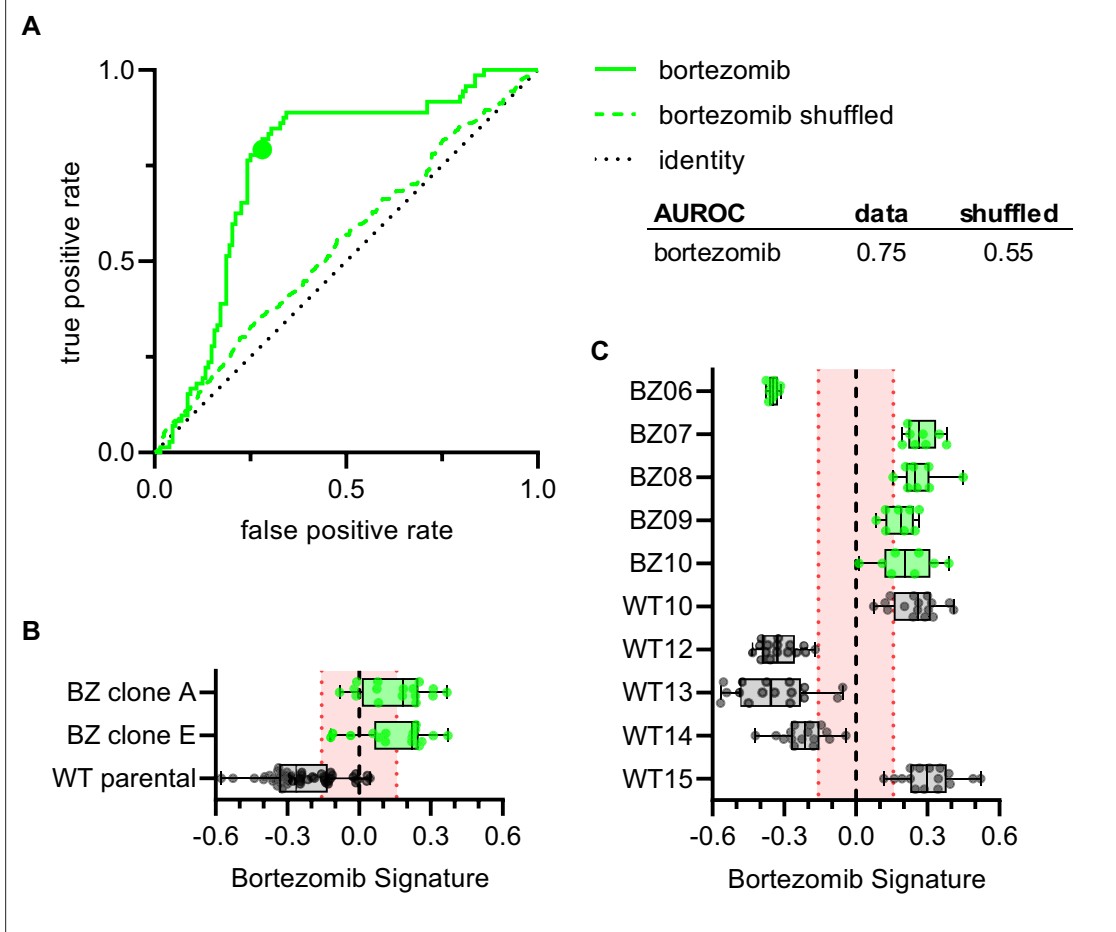

**Figure 5.** Bortezomib Signature correctly characterizes bortezomib sensitivity of seven out of ten clones not included in the training, validation, test, or holdout datasets. (**A**) ROC curve for the Bortezomib Signature of clones in (**B**) and (**C**) (solid line) and shuffled data (dashed line). Colored point is the corresponding false positive and true positive rate at the absolute minimum threshold. Black dashed line is the identity line where false positive rate = true positive rate. AUROC reported for the data and shuffled data. (**B**) Box plots of Bortezomib Signatures for bortezomib-resistant clones A and E (n = 16 profiles each) and HCT116 parental cells (n = 48 profiles). (**C**) Box plots of Bortezomib Signatures for bortezomib-sensitive clones WT10, WT12-15 (n = 16 profiles each) and bortezomib-resistant clones BZ06-10 (n = 8 profiles each). Plots show values for individual well profiles (points), range (error bars), 25th and 75th percentiles (box boundaries), and median. Dashed vertical black line is Bortezomib Signature = 0, dashed vertical red lines are the 95% confidence interval for Bortezomib Signatures of 1000 random permutations of the data. 4 independent experiments (biological replicates).

The online version of this article includes the following figure supplement(s) for figure 5:

**Figure supplement 1.** Examining the accuracy of clone classification and misclassification of clones.

ROC (AUROC) curve for ixazomib-resistant and CB-5083-resistant clones (0.63 and 0.60, respectively) was lower than those calculated for the training, validation, test, and holdout datasets for bortezomib-resistant clones. In addition, many of the Bortezomib Signatures for well profiles of ixazomib- and CB-5083-resistant clones, particularly those for CB-5083-resistant clones, landed within the 95% confidence interval of the randomly permuted data. These results suggest that the Bortezomib Signature is not a general signature of UPS-targeting drug resistance, nor of our cloning protocol, and instead has some specificity for bortezomib.

## Bortezomib Signature characterizes bortezomib sensitivity of clones not included in the training dataset

To examine whether the Bortezomib Signature could correctly characterize the bortezomib sensitivity of clones not included in the training, validation, holdout, or test datasets, we imaged bortezomib-sensitive (WT10, WT12-WT15) and bortezomib-resistant clones (BZ06-BZ10) using the Cell Painting protocol. The Bortezomib Signature had an AUROC of 0.75, compared to 0.55 for the shuffled data

(*Figure 5A*) and correctly characterized the bortezomib resistance of HCT116 parental cells and our bortezomib-resistant clones A and E, which we included as controls (*Figure 5B*). The Bortezomib Signature also correctly characterized the bortezomib resistance of four out of five bortezomib-resistant clones and three out of five bortezomib-sensitive clones not included in the training dataset (*Figure 5C*). In addition, the majority of Bortezomib Signatures for these well profiles landed outside the 95% confidence interval for the randomly permuted data. While the Bortezomib Signature correctly characterized the bortezomib sensitivity of most clones, it consistently misclassified others (WT10, WT15, and BZ06; *Figure 5—figure supplement 1A*). Proliferation assays conducted in earlier experiments showed that WT10 and WT15 were sensitive to bortezomib while BZ06 was resistant (*Figure 1—figure supplement 2A and B*). By comparing these incorrect predictions with high-confidence correct predictions, we observed differences that varied by clone type, suggesting unique morphology may be driving each of these misclassifications (*Figure 5—figure supplement 1B and C*). These results are consistent with the Bortezomib Signature being generalizable to clones not included in the training dataset and suggest that morphological profiling has the potential to identify bortezomib-resistant clones based on the morphological features of cells in the absence of drug treatment.

## Discussion

We used Cell Painting, a high-throughput, high-content image acquisition and analysis assay, as a target-independent method to capture the morphological profiles of cells that were either resistant or sensitive to the UPS-targeting anticancer drug, bortezomib, in the absence of drug treatment. After processing profiles to reduce the impact of technical variables, we generated a signature of bortezomib resistance and characterized the performance of this signature using machine learning best practices. This Bortezomib Signature correctly predicted the bortezomib resistance of seven out of ten clones not included in the training dataset and was more specific to bortezomib-resistance given its limited ability to identify clones that were resistant to other UPS-targeting drugs, ixazomib and CB-5083. All three drugs tested target the UPS, however bortezomib and ixazomib both bind the same subunit of the proteasome, albeit with potentially non-overlapping spectrums of off-targets (*Baggish et al., 2010*). These data suggest that the Bortezomib Signature may be specific to the drug bortezomib and not a general signature of resistance to UPS-targeting drugs. The Bortezomib Signature is conceptually similar to the on-disease/off-disease score (*Heiser et al., 2020*). Both require three phenotypic measurements: a target phenotype representing ideal, a disease phenotype, and a new phenotype to classify. However, our approach is technically different (non-parametric compared to linear projection) and our goals are different (phenotypic classification compared to perturbation alignment). Other methods also enable phenotype labeling, but they focus on single-sample annotation without regard to a target phenotype (*Nyffeler et al., 2020*; *Rohban et al., 2017*; *Simm et al., 2018*; *Wawer et al., 2014*). Our work demonstrates that there are morphological features of drug resistance in cells that can be identified using Cell Painting and provides a reproducible pipeline for generating morphological signatures of drug resistance.

The Bortezomib Signature's performance was not perfect; it misclassified three clones not included in the training dataset. Interestingly, one of the misclassified clones (BZ06) had reduced sensitivity to mitoxantrone as well as bortezomib. Given the considerable genetic heterogeneity in this mismatch repair-deficient HCT116 cell line (*Glaab and Tindall, 1997*; *Umar et al., 1994*), it is possible that some misclassified clones have orthogonal mechanisms of resistance or unrelated mutations contributing to their morphological profiles. Targeted sequencing of the PSMB5 proteasome subunit in bortezomib-resistant clones may provide information regarding origins of these misclassifications, as multiple mutations have been identified in bortezomib-resistant clones (*Wacker et al., 2012*). Determining the underlying reason for the misclassification of bortezomib-sensitive clones would require further studies.

Together, our work demonstrates the potential for morphological profiling with Cell Painting to be used as an unbiased method to characterize resistance in the absence of drug treatment. Our results indicate that different mechanisms of bortezomib resistance may generate distinct morphological profiles; with larger and broader training datasets, it may be possible to identify signatures for distinct mechanisms of bortezomib resistance as well as signatures of resistance to other drugs. Although it is unclear whether this method can be extended to patient samples, where identifying intrinsic drug

resistance in cells prior to treatment has the potential to improve targeted cancer therapy, our results are an encouraging proof of concept. We expect that further refinement may develop Cell Painting as a tool for identifying drug-resistant cells, perhaps even guiding strategies to overcome resistance.

# Materials and methods

**Key resources table**

| Reagent type (species) or resource | Designation | Source or reference | Identifiers | Additional information |
|---|---|---|---|---|
| Cell line (*Homo sapiens*) | HCT116 | ATCC | CCL-247; RRID:CVCL_0291 | |
| Chemical compound, drug | bortezomib | LC Laboratories | cat # 1408 | |
| Chemical compound, drug | taxol | Sigma | cat # T7402 | |
| Chemical compound, drug | mitoxantrone | TOCRIS | cat # 4250 | |
| Chemical compound, drug | ixazomib | ApexBio | cat # A4007 batch 2 | |
| Chemical compound, drug | CB-5083 | MedChemExpress | cat # HY-12861 batch 19554 | |
| Other | MitoTracker Deep Red | Invitrogen | cat # M22426 | |
| Other | Phalloidin AF568 | Invitrogen | cat # A12380 | |
| Other | Concanavalin A AF488 | Invitrogen | cat # C11252 | |
| Other | Hoechst 33342 | ThermoFisher | cat # 62249 | |
| Other | Hoechst 33342 | Invitrogen | cat # H3570 | |
| Other | Wheat-germ agglutinin AF555 | Invitrogen | cat # W32464 | |
| Other | SYTO14 Green | Invitrogen | cat # S7576 | |

## Cell culture

HCT116 cells (RRID: CVCL_0291) were purchased from ATCC (CCL-247) and were maintained in McCoy's 5 A Medium (Gibco) supplemented with 10% (v/v) FBS (Sigma) and cultured at 5% $CO_2$ and 37 °C. Cells were determined to be mycoplasma-free using a PCR-based method (*Uphoff and Drexler, 2013*). Bortezomib-resistant, ixazomib-resistant, and CB-5083-resistant clones were isolated as previously described (*Wacker et al., 2012*). Briefly, HCT116 cells were plated in 150 mm dishes and grown in the presence of the desired drug at a concentration that resulted in the death of the majority of cells (selection concentrations: bortezomib [LC Laboratories], 12 nM; ixazomib [ApexBio], 150 nM; CB-5083 [MedChemExpress], 600 and 700 nM). The locations of single surviving cells were identified using brightfield microscopy and marked on the plate. Cells were allowed to expand into colonies over 2–4 weeks and colonies were isolated using cloning rings. Bortezomib-sensitive clones were generated by diluting HCT116 cells into 96-well plates and wells containing single cells as identified by brightfield microscopy were marked. Colonies that grew in these marked wells were expanded and used for experiments. Bortezomib-resistant clones A and E were provided by the Kapoor laboratory having been previously published (*Wacker et al., 2012*).

## Proliferation assays

Cell proliferation was evaluated using an Alamar Blue assay (*O'Brien et al., 2000*). Briefly, cells were plated in duplicate or triplicate in sterile 96-well Clear Microplates (Falcon) under described culture conditions, with 1000 cells in 100 µL per well and allowed to adhere overnight. After cells attached to the plate, 50 µL of media containing drug (bortezomib, ixazomib, CB-5083, taxol [Sigma], or mitoxantrone [TOCRIS]) was added to each well. The final DMSO concentration was 0.1% for all wells, including three wells with media only as background measurements. Plates were incubated for 72 hr at 5% $CO_2$ and 37 °C before adding Alamar Blue (resazurin sodium salt, final concentration 50 µM). Cells were incubated with Alamar Blue for 3–4 hr and then imaged with a Synergy Neo plate reader using excitation: 550 nm and emission: 590 nm (Agilent). The average plate background (media only with 0.1% DMSO) was subtracted from the average fluorescence for each condition and the resulting value was normalized by dividing by the background-subtracted value for each condition's control (cells treated

with 0.1% DMSO). Using the data from our proliferation assays, we calculated the median lethal dose (LD50) for each of our drugs of interest by fitting data of normalized growth vs. log[drug concentration] to a sigmoidal dose-response curve using GraphPad Prism (v.9.2.0) (*Figure 1—source data 1*).

## Cell Painting

High-throughput imaging was performed according to the published Cell Painting protocol (*Bray et al., 2016*). HCT116 cells were plated at concentrations of 2.5 or $5x10^3$ cells/mL in 96-well glass-bottomed tissue culture dishes (Greiner Bio-One) and allowed to adhere for 48–72 hr prior to fixation. At either 4 or 13 hr prior to fixation, cells were treated with either 0.1% DMSO or 7 nM bortezomib and 30 min prior to fixation cells were treated with MitoTracker Deep Red (500 nM, Invitrogen). 16% paraformaldehyde (EMS) was added to each well for a final concentration of 3.2% and cells were fixed in the dark at room temperature for 20 min. Wells were washed with HBSS (Invitrogen), permeabilized with 0.1% Triton-X for 15 min, and then washed twice with HBSS before incubating with staining solution (5 U/mL phalloidin AF568 [Invitrogen], 100 µg/mL concanavalin A AF488 [Invitrogen], 5 µg/mL Hoechst 33342 [ThermoFisher or Invitrogen], 1.5 µg/mL wheat-germ agglutinin AF555 [Invitrogen], 3 µM SYTO14 Green [Invitrogen], and 1% bovine serum albumin [BioWorld] in HBSS) in the dark for 30 minutes. Wells were then washed twice with HBSS and imaged using an ImageXpress high-content imaging system (Molecular Devices) with a 20x0.45 NA S Plan Fluor ELWD objective (Nikon) and captured with a Zyla 5.5 sCMOS detector (Andor Technology). Each well was imaged at 12–17 non-overlapping sites in five channels using Semrock filters (mito: Cy5-4040B-NTE-ZERO, AGP: TxRed-4040C-NTE-ZERO, RNA: Cy3-4040C-NTE-ZERO, ER: FITC-3540C-NTE-ZERO, and DNA: DAPI-5060C-NTE-ZERO).

## Image data processing

We used CellProfiler versions 3.1.8 and 3.1.9 (*Caicedo et al., 2018*) to perform the standard processing pipeline of illumination correction, single-cell segmentation, and morphology feature extraction. We performed per-plate illumination correction to adjust for uneven background intensity that commonly impacts microscopy images. We also developed per-plate analysis pipelines for single-cell segmentation and feature extraction. We extracted 3,528 total cell morphology features from all 25,331,572 cells we captured in this experiment. The 3528 features represent stain intensities, stain co-localization, textures, areas, and other patterns extracted from all five imaging channels and different segmentation objects (nuclei, cytoplasm, total cells). Feature details are described in the documentation for CellProfiler (https://cellprofiler-manual.s3.amazonaws.com/CellProfiler-3.1.9/help/output_measurements.html). We include all image analysis pipelines at https://github.com/broadinstitute/profiling-resistance-mechanisms (*Way et al., 2023*).

Following feature extraction, we applied an image-based analysis pipeline to generate our final analytical set of treatment profiles (*Caicedo et al., 2017*). We first used cytominer-database to ingest all single-cell, per-compartment CellProfiler output files (comma separated) to clean column names, evaluate integrity of CellProfiler output CSVs, and output single-cell SQLite files for downstream processing. Next, we used pycytominer (github hash c1aa34b641b4e07eb5cbd424166f31355abdbd4d) for all image-based profiling pipeline steps. In the first step, we median aggregated all single cells to form well-level profiles (*Way et al., 2022*). Next, we performed a step called annotation, which merges the consistent platemap metadata with the well-level profiles. Third, we performed standard z-score normalization to ensure all features are measured on the same scale with zero mean and unit variance. Lastly, we performed feature selection, which removed features with low variance, high correlation (>0.9 Pearson correlation), features with missing values, features on our blocklist (*Way, 2020*), and features with outliers greater than 15 standard deviations, which we suspected were measured in error. For developing our final analytical datasets (see section, Constructing the resistance signature) we performed normalization within each plate but performed a combined feature selection across all plates per analytical dataset using the same procedures described previously, which resulted in 782 features. We applied the same pipeline uniformly across all plates. We did not detect large differences in variance that could be attributed to well position and batch and therefore did not apply batch effect correction. Our full image data processing pipeline is publicly available at https://github.com/broadinstitute/profiling-resistance-mechanisms (*Way et al., 2023*).

## Constructing the resistance signature

After processing all images and forming normalized and feature selected profiles per well, we performed additional analyses to explore the results and discover a morphology profile of bortezomib resistance. We performed initial comparisons of morphological profiles using Morpheus (https://software.broadinstitute.org/morpheus) to create similarity matrix heatmaps.

We aimed to discover a generalizable signature of bortezomib resistance from the normalized profiles. Our approach was to identify features that were significantly different by resistance status and not significantly impacted by technical covariates. To do so, we carefully constructed datasets for training and evaluating signature performance (*Figure 3—source data 1*). To generate our training dataset, we selected a set of six plates consisting of five bortezomib-sensitive and five bortezomib-resistant clones that we collected on three different days, which showed high within-replicate reproducibility (technical replicates; data not shown). A seventh plate was held-out from signature generation in order to analyze generalizability between plates (holdout dataset). We evaluated the signature in five scenarios: (1) clones held-out on the same plates used to generate the training dataset (validation dataset, *Figure 3B*), (2) HCT116 parental cells and clones with PSMB5 mutations known to confer resistance to bortezomib (test dataset, *Figure 3C*; *Wacker et al., 2012*), (3) clones held-out on a separate plate (holdout dataset, *Figure 3D*), (4) clones selected to be resistant to other drugs (ixazomib and CB-5083, *Figure 4*), and (5) bortezomib-resistant clones not included in the training dataset (*Figure 5*). All cells on these plates were incubated with 0.1% DMSO for either 4 or 13 hours.

Using data from the ten clones in our training dataset (20–21 technical replicates per clone, see *Figure 3—source data 1*), we fit two linear models for all 782 CellProfiler features (post normalization and feature selection) to discover features that varied strongly with technical variants (batch, cell count, incubation time, or clone ID) and features that varied strongly with resistance status (bortezomib-sensitive or bortezomib-resistant). In the first linear model, we quantified the per feature variance contribution of resistance status ($\beta_{resistance\ status}$), batch ($\beta_{batch}$), incubation time ($\beta_{incubation\ time}$), and clone ($\beta_{clone\ ID}$) to each CellProfiler feature ($Y_j$) where $\varepsilon$ is the error term:

$$Y_j = \beta_{intercept} + \beta_{resistance\ status}X_{resistance\ status} + \beta_{batch}X_{batch} + \beta_{incubation\ time}X_{incubation\ time} + \beta_{clone\ ID}X_{clone\ ID} + \varepsilon$$

Fitting this model produced a goodness of fit $R^2$ value per feature and individual beta coefficients per covariate. Furthermore, we calculated a Tukey's Honestly Significant Difference (Tukey's HSD) post hoc test per model to determine which categorical covariate comparison contributed to a significant finding and to control for within-covariate-group multiple comparisons through a family-wise error rate (FWER) adjustment that accounts for different within-group sizes (e.g. three different batches in the comparison, two different resistance statuses, etc.) (*Tukey, 1949*).

Separately, we fit another linear model on continuous features to adjust for features that were significantly impacted by well confluence ($\beta_{cell\ count}$) as it is expected that dense wells will impact certain morphology features, which we want to avoid in the resistance signature:

$$Y_j = \beta_{intercept} + \beta_{resistance\ status}X_{resistance\ status} + \beta_{cell\ count}X_{cell\ count} + \varepsilon$$

By fitting these models, we quantified the variance contribution of four technical covariates (incubation time, batch, clone ID, and cell count) and our biological variable of interest (resistance status), and, based on the first linear model, we have knowledge of which group comparisons were significant in each category (via Tukey's HSD). We further refined the signature by filtering features that did not pass a Bonferonni adjusted alpha threshold calculated across all 782 features ($0.05/782 = 6.4\times10^{-6}$).

We next applied an exclusion criterion to isolate features that contributed to resistance status. We excluded features that were significantly different across incubation times, batches, and cell counts. We also excluded features that were different within clone type (features varying between two or more pairs of bortezomib-sensitive clones) to reduce the contribution of features that may mark generic inter-clone differences nonspecific to resistance status. This procedure resulted in a total of 45 features that were significantly different by resistance status and not significantly impacted by any of the technical covariates we considered. Of the 45 features, 14 had higher values in resistant clones and 31 had lower values in resistant clones (*Figure 2—source data 1*).

We were also interested in comparing the ability of different feature spaces to cluster clones of the same type (resistant vs. sensitive). This analysis would determine if the Bortezomib Signature features, which we derived using linear modeling to isolate biological from technical variables, had a greater ability to cluster. We compared the Bortezomib Signature against three other feature spaces: (1) the full feature space, (2) standard feature selection (see Image data processing methods), and (3) 45 randomly selected features. We performed two analyses using these four feature spaces including Uniform Manifold Approximation and Projection (UMAP) (*McInnes et al., 2018*) and k-means clustering. For UMAP, we used default umap-learn parameters to identify two UMAP coordinates per feature space. We then visualized the clusters by their resistance status and Bortezomib Signature score. The UMAP analysis represents a qualitative analysis. Next, we applied k-means clustering with 25 initializations across a range of 2–14 clusters (k). Prior to clustering and for each feature space, we applied principal component analysis (PCA) and transformed each feature space into 30 principal components. This step was necessary to compare k-means clustering metrics, which are sensitive to the feature space dimensionality. We applied a Fisher's exact test to each cluster using a two-by-two contingency matrix that specified cluster membership for each clone classification (resistant vs. sensitive). We visualized the mean odds ratio and max cluster odds ratio for each feature space across k. A high odds ratio tells us that the feature space effectively clusters clones of the same resistance status. Lastly, we calculated Silhouette width (the average proximity between samples in one cluster to the second nearest cluster) for each feature space across k.

## Applying the signature

We used the *singscore* method (*Foroutan et al., 2018*) to characterize individual profiles of different clones as either bortezomib-resistant or bortezomib-sensitive. *Singscore* is a rank-based method that was originally developed to analyze the direction and significance of previously defined molecular signatures on transcriptomic data. The method calculates a two-part signature for each direction list (14 up and 31 down) and calculates an internal rank per profile of how highly ranked and lowly ranked each of the up and down features are, respectively. The method then adds the up and down rank scores to form a total *singscore* per sample, which ranges between –1 and 1 and represents a rank-based normalized concordance score that can be directly compared across profiles that may have been normalized differently. Therefore, the score is robust to outliers and different normalization procedures. In addition to calculating the *singscore* per sample, we also calculated *singscore* with 1,000 random permutations, in which we randomly shuffled feature rankings to derive a range in which a sample may be scored by chance. Some profiles were consistently predicted incorrectly with high confidence but in the opposite direction (see *Figure 5—figure supplement 1*). For a well-level profile to be categorized as high-confidence (in either the correct or incorrect directions), it needed to score beyond the 95% confidence interval of the randomly permuted data range. For example, a high-confidence incorrect resistant profile would have a Bortezomib Signature below 95% confidence interval of the randomly permuted data. To evaluate the features driving the differences in these samples, we applied two-sample Kolmogorov–Smirnov (KS) tests per Bortezomib Signature feature. We applied these tests to two separate groups: (1) misclassified bortezomib-sensitive vs. high-confidence accurate bortezomib-sensitive and (2) misclassified bortezomib-resistant vs. high-confidence accurate bortezomib-resistant.

## Signature evaluation

We used several metrics to evaluate signature quality across five different evaluation scenarios (validation, test, holdout, other UPS-targeting drugs, and clones not included in the training dataset [biological replicates]). Because we are measuring a binary decision in a balanced dataset (roughly the same amount of positive as negative classes), we used accuracy (total correct / total chances) to quantify performance. We also calculated mean average precision using sci-kit learn, averaging over samples along the precision recall curve (*Varoquaux et al., 2011*), which is a measure of separation between the two resistance classes (higher being more separation). We also calculated receiver operating characteristic (ROC) curves and area under the ROC curve (AUROC) using sci-kit learn. AUROC compares the ability to distinguish positive samples across signatures.

## Acknowledgements

The authors gratefully acknowledge funding from the Starr Cancer Consortium (112-0039 to TMK and AEC), the National Institutes of Health (NIH MIRA R35 GM122547 to AEC, NIH MIRA R35 GM130234 to TMK, NIH NRSA T32 GM066699 to MEK, NIH T32 GM115327 Chemistry-Biology Interface Training Grant to the Tri-Institutional PhD Program in Chemical Biology to AYB), and the National Science Foundation (NSF GRFP 2019272977 to AYB). We thank Erin Weisbart for uploading image data to the public database. We followed the International Committee of Medical Journal Editors (ICMJE) recommendations for authors and contributors.

## Additional information

### Funding

| Funder | Grant reference number | Author |
|---|---|---|
| Starr Cancer Consortium | 112-0039 | Anne E Carpenter |
| National Institutes of Health | R35 GM122547 | Anne E Carpenter |
| National Institutes of Health | R35 GM130234 | Tarun M Kapoor |
| National Institutes of Health | T32 GM066699 | Megan E Kelley |
| National Institutes of Health | T32 GM115327 | Adi Y Berman |
| National Science Foundation | NSF GRFP 2019272977 | Adi Y Berman |

The funders had no role in study design, data collection and interpretation, or the decision to submit the work for publication.

### Author contributions

Megan E Kelley, Conceptualization, Data curation, Formal analysis, Validation, Investigation, Visualization, Methodology, Writing – original draft, Writing – review and editing; Adi Y Berman, Conceptualization, Data curation, Formal analysis, Validation, Investigation, Methodology, Writing – review and editing; David R Stirling, Data curation, Formal analysis, Validation, Investigation, Methodology, Writing – review and editing; Beth A Cimini, Data curation, Software, Investigation, Writing – review and editing; Yu Han, Software, Formal analysis, Writing – review and editing; Shantanu Singh, Conceptualization, Resources, Supervision, Investigation, Project administration, Writing – review and editing; Anne E Carpenter, Tarun M Kapoor, Conceptualization, Resources, Supervision, Funding acquisition, Investigation, Project administration, Writing – review and editing; Gregory P Way, Conceptualization, Data curation, Software, Formal analysis, Supervision, Validation, Investigation, Visualization, Methodology, Writing – original draft, Project administration, Writing – review and editing

### Author ORCIDs

Megan E Kelley ⓘ http://orcid.org/0000-0002-0251-5054
Adi Y Berman ⓘ http://orcid.org/0000-0002-1460-7562
David R Stirling ⓘ http://orcid.org/0000-0001-6802-4103
Beth A Cimini ⓘ http://orcid.org/0000-0001-9640-9318
Yu Han ⓘ http://orcid.org/0000-0001-5507-9228
Shantanu Singh ⓘ https://orcid.org/0000-0003-3150-3025
Anne E Carpenter ⓘ http://orcid.org/0000-0003-1555-8261
Tarun M Kapoor ⓘ http://orcid.org/0000-0003-0628-211X
Gregory P Way ⓘ https://orcid.org/0000-0002-0503-9348

### Decision letter and Author response

Decision letter https://doi.org/10.7554/eLife.91362.sa1
Author response https://doi.org/10.7554/eLife.91362.sa2

## Additional files

### Supplementary files
• MDAR checklist

### Data availability
All data generated during this study are provided in the dataset cpg0028-kelley-resistance, available in the Cell Painting Gallery on the Registry of Open Data on AWS (https://registry.opendata.aws/cellpainting-gallery/). Processed data, source data files, and code to reproduce this analysis are available at https://github.com/broadinstitute/profiling-resistance-mechanisms (*Way et al., 2023*).

The following dataset was generated:

| Author(s) | Year | Dataset title | Dataset URL | Database and Identifier |
|---|---|---|---|---|
| Kelley ME, Berman AY, Stirling DR, Cimini BA, Han Y, Singh S, Carpenter AE, Kapoor TM, Way GP | 2023 | Cell Painting images to produce a high-content phenotypic signature of Bortezomib resistance | https://registry.opendata.aws/cellpainting-gallery/ | Registry of Open Data on AWS, cpg0028-kelley-resistance/ |

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
