## [Editor Report]

In this work, the authors have profiled the morphological signatures of the HCT116 cell line and correlated them with bortezomib treatment response, which could provide novel insight into the research of resistance in multiple myeloma from the perspective of morphology. The findings are supported by solid evidence and sufficient experimental validation.

---

## [Decision Letter]

[Editors' note: this paper was reviewed by Review Commons.]

---

## [Author Response]

*General statements*

We would like to thank the editorial staff and the reviewers for their handling of our manuscript. We were very pleased with the timely communications from Review Commons, and we are grateful to have been assigned this insightful and constructive group of reviewers.

The reviewers were well-suited to evaluate our work based on their stated areas of expertise (cancer biology, image analysis, machine learning, cell-based screening, etc.). As such, we received thoughtful and constructive feedback, which we have already incorporated into our attached revision. We are confident that these reviews have improved our manuscript.

Our goal with this manuscript is to present a proof-of-concept study where high-content imaging and morphological profiling are used to characterize drug resistance in clonal cell lines. The main criticism from reviewers was that our original manuscript may have overstated our method’s ability to discriminate the signal of bortezomib resistance and that any extension beyond cultured cells (to patient samples for example) would require significant follow-up studies. The reviewers suggested that such work would be beyond the scope of our study, and recommended toning down our language to better reflect the limitations of this proof-of-concept work. We have embraced this suggestion, extensively revising our text, and we now believe our language and tone more accurately reflects our results. The reviewers also suggested follow-up computational analyses to more robustly characterize the bortezomib resistance signature. We have performed these analyses and added their description to our revised manuscript. We feel that these analyses have improved understanding of the signature, and will help a reader to gain a deeper understanding of our results and methodology.

The reviewers also suggested several minor changes; many of which we embraced fully, but others that we chose not to incorporate. We felt that a lack of clarity in our text contributed to these reviewer suggestions. In these cases, we improved clarity in the text and responded to each comment point-by-point in the “prefer not to carry out” section. Further, we address all reviewer comments in the following document point-by-point, grouped by common themes across reviewers (e.g., tone, clarity, analyses, etc.).

Lastly, a common theme among reviewer comments was their appreciation for our strong methodology and data transparency (examples pasted below). We are extremely gratified by this observation as we feel this is a particular strength of our manuscript. In addition, we were pleased to see reviewers engaged by our work, acknowledging the interest this manuscript is likely to generate among a broad range of scientific disciplines.

Examples of reviewer appreciation of our strong methodology and data transparency:

Reviewer 1: “However, this does not imply that the same approach can not achieve the goal, perhaps by using other cell painting markers for bortezomib-sensitivity, or with the same markers to assess sensitivity of different drugs. The cell painting + analysis approaches are not new and the clinical impact is questionable, but the technical aspects (data, analysis) are exceptional and the concept may hold as I described above.”

Reviewer 2: **“**The paper is well written, and the text is clear, as is the presentation of data and transparency of methods being utilized. The methods were applied appropriately and followed established standards in the field. The paper's premise is timely and interesting, addressing a pressing issue in cancer therapy: making informed treatment decisions fast, based on markers found in tumors early in tumor development, and using image-based screening for characterizing drug resistance before treatment could be an option. A fascinating bit of the manuscript is the description of the feature selection from the screen is done systematically, considering the technical and biological variability and technical artifacts and modeling covariates using linear models seems a very appropriate way of doing so and could serve as another proof of concept that this is indeed the most robust way of modeling and removing signal of technical covariates from the data.”

Reviewer 3: “The strengths of this study are the machine learning best practice and detailed methodology. The experiments could be reproduced and statistical analysis is more than adequate. The analysis takes into account batch effects, well position, differences in cell numbers, and other sources of technical variation that complicate high-content image analysis. It is a good exemplar of how unsupervised morphological profiling can be applied to imaging data. The major limitation is the generalizability of this particular method for patient samples. This could be addressed in the Discussion.”

*Description of the revisions*

We have incorporated all planned revisions.

*Description of the revisions that have already been incorporated in the transferred manuscript*

Text revisions already carried out

[Text revision] We have materially toned down our claims in the manuscript in two distinct areas: (A) model performance and (B) potential clinical application.

Model performance. We specifically balanced our discussion of the discriminative signal of the Bortezomib Signature. While the signature adequately separated never-before-seen wildtype and resistant clones with metrics well above randomly permuted baselines (accuracy near 80%, average precision about 70%, area under the ROC curve (AUROC) about 84%), there were many limitations that we should have more explicitly highlighted. For example, many individual profiles were incorrectly classified, some clones were predicted entirely incorrectly, and many profiles did not receive Bortezomib Signature scores above the randomly permuted baseline. We have more clearly discussed these limitations and used more balanced language (see key examples of text-based changes below). Additionally, we modified a figure (now Figure 3) to include boxplots of clones that explicitly show the Bortezomib Signature scores of each well profile and permit examination of the strength of the signature for each clone (previously found in Figure 2-Supplement 9). Lastly, we add a new supplementary figure (now Figure 5-Supplement 1) that describes a feature space analysis of misclassified samples. Please note that this figure rearrangement and new analysis helped to balance our claims, but were also performed in response to other tangential reviewer comments.Clinical application. In the abstract, introduction, and discussion, we further emphasized that this work is a proof of concept, and that more advances must be made prior to clinical application.

We made these changes in direct response to the following reviewer comments:

Reviewer 1 - Major Comment 1 (relevant excerpts)While I am convinced that the signature captures morphological phenotypes associated with drug resistance, at the cumulative scale, the discriminative signal of a single cell type seems weak… With Fig. 4, the data fully supports the argument that the bortezomib-signature encodes bortezomib-resistance, but the signal is weak. Thus statements such as "We found the Bortezomib Signature could predict whether a cell line was bortezomib-resistant or bortezomib-sensitive" (line #172) and the specificity statements in the abstract" (line #28) are not supported by the data in my opinion. I would recommend the authors to tune down these and other related statements throughout the manuscript.Reviewer cross-commenting - Reviewer 1My main critic is regarding "over selling" a weak discriminative signal. Specifically, I am not convinced that the major claims regarding predicting sensitivity and specificity at the single cell types scales are supported by the data. Since reviewer #2 and #3 did not raise this concern I think it is worth discussion here.Once these statements are tuned down - I think no significant additional work is needed to make the point that they can measure a discriminative signal. If they want to make these claims, perhaps they'd like to collect more data to gain statistical power (but I am not optimistic this will work at the single cell level).Personally, I was happy with the authors' choice of cell lines not included in the training dataset. I am not convinced that additional cell lines + validations are necessary for making the point of a proof of principle.Reviewer cross-commenting - Reviewer 2I agree that, perhaps, my major criticism of the paper was the manuscript's 'overselling' of claims that were only weakly supported by the data. Yes, if the authors tune down their claims and clearly state that this is an interesting starting point and proof of concept study, it might be ok to publish with only minor revisions. If the claims should be more generalized, then this study needs more data supporting the conclusions and the method's predictive power.Reviewer 2 - Major Comment 8Lastly, I find some misfits between the question, the model used, and the conclusions drawn. The authors start by exploring the problem of bortezomib resistance in cancer treatment, which they say is a devastating issue for patients with, e.g., multiple myeloma. Yet, the authors use HCT116 as their model cell line, a microsatellite instable, colorectal cell line with several intrinsic mutations that make it a difficult model to address physiologically relevant medical problems after all. The authors then go on to suppose that their method might be suitable to diagnose resistance in patient samples, but I am not convinced this conclusion can be speculated based on data from HCT cells. I suggest the authors test their approach on at least two other cell lines (maybe from different tissues) and benchmark their results against a dataset of digital pathology where such predictions are made from stained and analyzed tissue slices. This way, after a thorough benchmark against related third-party data sets, the method would significantly gain relevance, the paper would appeal to a broader audience, and the advance gains more merit.Reviewer 3 - Major Comment 5It is not clear from the Discussion whether this type of analysis is more broadly applicable to cell lines derived from patients, rather than selected from a parental cell line, or if this approach would be more efficient than genotyping or next-gen sequencing. How many replicates and ground truth cell lines would be necessary for predictive confidence?

We edited the last two sentences of the abstract to tone down specificity claims (“provide evidence”) and clarify that we are establishing a “proof-of-concept framework”.

“This signature predicted bortezomib resistance better than resistance to other drugs targeting the ubiquitin-proteasome system. Our results establish a proof-of-concept framework for the unbiased analysis of drug resistance using high-content microscopy of cancer cells, in the absence of drug treatment.”

We revised the last paragraph of the introduction to contrast bortezomib predictions with ixazomib/CB-5083 predictions, and to remove claims about “using microscopy to guide therapy”.

“This morphological signature correctly predicted the bortezomib resistance of seven out of ten clones not included in the signature training dataset. Overall, our results establish a proof-of-concept framework for identifying unbiased signatures of drug resistance using high-content microscopy. The ability to identify drug-resistant cells based on morphological features provides a valuable orthogonal method for characterizing resistance in the absence of drug treatment.”

To tone down claims in the figures, we added boxplots to Figure 3 (previous Figure 2) showing specific distribution of signature scores per well profile and updated Figure 4 legend (previous Figure 3).

“Figure 4. Bortezomib Signature has limited ability to characterize clones resistant to other ubiquitin-proteasome system inhibitors.”

We modify the following text in the discussion to tone down claims of specificity and clinical utility:

“This Bortezomib Signature correctly predicted the bortezomib resistance of seven out of ten clones not included in the training dataset and was more specific to bortezomib-resistance given its limited ability to identify clones that were resistant to other UPS-targeting drugs.

Though it is unclear whether this method can be extended to patient samples, where identifying intrinsic drug resistance in cells prior to treatment has the potential to improve targeted cancer therapy, our results are an encouraging proof of concept. We expect that further refinement may develop Cell Painting as a tool for identifying drug-resistant cells, perhaps even guiding strategies to overcome intrinsic resistance.”

[Text revision] We defined LD50 in text (originally line #97), changed description of resistant clone selection to remove main text references to LD90 (originally line #87), and stated drug concentrations used for selection in Methods. We also defined LD90 in the Methods and described its role in determining the drug concentrations to use for clone selection. This change was in response to the following comments:

Reviewer 1 - Minor Comment 2What is LD90 (line #87)? LD50 (line #97)?Reviewer 2 - Minor Comment 5What was the LD 90 per drug on HCT cells? Rather than LD90 foldchanges, absolute concentrations should be used in the results and discussion to allow the reader to vet the conclusions.

“To determine the appropriate drug concentrations to use in order to isolate drug-resistant clones, we performed proliferation assays on HCT116 parental cells with our drugs of interest: bortezomib (proteasome inhibitor), ixazomib (proteasome inhibitor), or CB-5083 (p97 inhibitor) (Fig. 1-Supplement 1 A-D).”

“We characterized the bortezomib-resistant clones and found that the median lethal doses (LD50s) were ~2.8- to ~9-fold that of HCT116 parental cells (Fig. 1-Supplement 2 B).”

“Briefly, HCT116 cells were plated in 150 mm dishes and grown in the presence of the desired drug at a concentration that resulted in the death of the majority of cells (selection concentrations: bortezomib, 12 nM; ixazomib, 150 nM; CB-5083, 600 and 700 nM).”

“Using the data from our proliferation assays, we calculated the median lethal dose (LD50) for each of our drugs of interest by fitting data of normalized growth vs. log[drug concentration] to a sigmoidal dose-response curve using GraphPad Prism (v.9.2.0) (Fig. 1-Supplement 1 D).”

[Text revision] We thank the reviewer for allowing us an opportunity to improve clarity on the clones we used. We now describe the total number of clones generated and removed unnecessary references to specific clones for ease of reading (originally lines #96-98) (We maintain all references to specific clones in the figures, legends, supplement, and methods)

Reviewer 1 - Minor Comment 3It was not clear to me in the text which and how many cell lines were evaluated and the reader is forced to go to the SI. For example, "(BZ01-10 and BZ clones A and E)" (line #96-97) and "wild-type clones (WT01-05, 10, and 12-15)" (line #98) appeared when presenting the results without a clear explanation and made it harder for me to follow. Summary of the data (for example, based on Figure 2-Supplement 8) can be briefly mentioned in the text to make it more clear for the reader.

We added the following to the second paragraph of the results:

“Together these methods provided a total of twelve bortezomib-resistant, five ixazomib-resistant, five CB-5083-resistant, and twelve bortezomib-sensitive clones as well as HCT116 parental cells for our experiments.”

[Text revision] We removed duplicate text (originally lines #115-125).

Reviewer 1 - Minor Comment 5Lines #104-111 were duplicated in lines #114-122.Reviewer 3 - Minor Comment 4Ten lines of text are duplicated on page 5.Reviewer 2 - Minor Comment 4 on page 5, paragraph 4, there is a sizeable copy-and-paste error of text being identically replicated.

[Text revision] We provided more intuition of the Bortezomib Signature in the results section (originally lines #150-151).

Reviewer 1 - Minor Comment 6The "Bortezomib Signature" is a critical measurement but is only briefly mentioned in lines 150-151 ("...based on the direction-sensitive ranking method for phenotype analysis, singscore (Foroutan et al., 2018)"). Please provide more information/intuition.

“We used these 45 features to compute a rank-based resistance score or “Bortezomib Signature” for each well profile based on the direction-sensitive method called singscore (Foroutan et al. 2018). Singscore ranks these 45 resistance-related features on a per sample basis and calculates a normalized score between -1 and 1, with higher values expected for bortezomib-resistant clones and lower values expected for bortezomib-sensitive clones.”

[Text revision] We clarified that DNA sequencing had been performed solely on clones A and E in a previous study (originally lines #88-90). Furthermore, one of the strengths of our approach is that it can identify resistant clones in an unbiased fashion prior to molecular characterization. It is beyond scope to perform these sequencing studies in the present paper.

Reviewer 2 - Minor Comment 3The authors talk about validating the mutation - PSMB5 by RNA-seq. However, the data for the genotyping/sequencing/characterization of these newly generated BZ-resistant lines are missing.

In the results, we clarify DNA sequencing that was previously performed on clones A and E

“We also isolated bortezomib-sensitive (wild-type; WT) clones by dilution of the HCT116 parental cell line and acquired two bortezomib-resistant clones (BZ clones A and E) both with mutations in PSMB5 identified by RNA sequencing performed in previous work (Fig. 1-Supplement 1 E) (Wacker et al. 2012).”

In the last paragraph of the discussion, we highlight the strength of our unbiased approach.

“Together, our work has demonstrated the potential for morphological profiling with Cell Painting to be used as an unbiased method to characterize resistance in the absence of drug treatment. Our results indicate that different mechanisms of bortezomib resistance may generate distinct morphological profiles; with larger and broader training datasets, it may be possible to identify signatures for distinct mechanisms of bortezomib resistance as well as signatures of resistance to other drugs. Though it is unclear whether this method can be extended to patient samples, where identifying intrinsic drug resistance in cells prior to treatment has the potential to improve targeted cancer therapy, our results are an encouraging proof of concept. We expect that further refinement may develop Cell Painting as a tool for identifying drug-resistant cells, perhaps even guiding strategies to overcome intrinsic resistance.”

[Text revision] We thank the reviewers for their suggestions. We agree that the description of the experimental design was somewhat unclear and have provided greater detail and clarity, particularly regarding the generation of clones. We used the HCT116 parental cell line to generate drug-resistant clones by identifying single surviving cells after drug treatment and allowing these cells to expand prior to isolating colonies for experimentation. We did not perform experiments to confirm whether these “clones” were isogenic and can not exclude cell migration during expansion or genetic drift as convoluting factors. However, we have provided greater detail in the descriptions of our method for clone isolation in order to address this concern.

Reviewer 1 - Minor Comment 1More information in Fig. 1's legend would be helpful to follow the experimental design. I found it hard to follow in its current form and had to go back to carefully reading the main text to fully understand.Reviewer 2 - Minor Comment 6The description of the resistant clonal populations is confusing. As I understand, no single-cell clones were isolated during the selection procedure. Thus, the training lines are not yet isogenic clones but oligoclonal sub-populations of the parental cell line. The authors could provide more details here and discuss the different characteristics of their sub-populations, e.g., their growth kinetics or molecular alterations.

We bolstered the description in the results.

“We first isolated and characterized drug-resistant cells (Fig. 1 A). To isolate drug-resistant clones, we used an approach we have described previously (Wacker et al. 2012; Kasap, Elemento, and Kapoor 2014) and the HCT116 cell line. These cancer cells express multidrug resistance pumps at low levels and are mismatch repair deficient, providing a genetically heterogeneous polyclonal population of cells (Umar et al. 1994; Papadopoulos et al. 1994; Teraishi et al. 2005) allowing for isolation of drug-resistant clones in 2-3 weeks. We hypothesize that a rapid selection of resistance could favor the isolation of clones with intrinsic resistance. To determine the appropriate drug concentrations to use in order to isolate drug-resistant clones, we performed proliferation assays on HCT116 parental cells with our drugs of interest: bortezomib, ixazomib, or CB-5083 (Fig. 1-Supplement 1 A-D). We also isolated bortezomib-sensitive (wild-type; WT) clones by dilution of the HCT116 parental cell line and acquired two published bortezomib-resistant clones (BZ clones A and E) both with mutations in PSMB5 identified by RNA sequencing performed in previous work (Fig. 1-Supplement 1 E) (Wacker et al. 2012). We characterized the bortezomib-resistant clones and found that the median lethal doses (LD50s) for bortezomib were ~2.8- to ~9-fold that of HCT116 parental cells (Fig. 1-Supplement 2 B). In contrast, bortezomib-sensitive clones had LD50s for bortezomib that ranged from ~0.7- to ~1.2-fold that of HCT116 parental cells (Fig. 1-Supplement 2 A). Together these methods provided a total of twelve bortezomib-resistant, five ixazomib-resistant, five CB-5083-resistant, and twelve bortezomib-sensitive clones as well as HCT116 parental cells for our experiments.”

We also updated the legend for Figure 1A.

“Figure 1. Experimental design for using Cell Painting to examine morphological profiles of drug-resistant cells. (A) Graphic of the experimental workflow: we isolated drug-resistant clones by treating parental HCT116 cells with a high dose of the desired drug and then expanded them for experiments. We isolated drug-sensitive clones by diluting HCT116 cells and then expanded them for experiments. We then performed proliferation assays on select clones to screen for multidrug resistance. Next, we performed Cell Painting on both drug-resistant and -sensitive clones, using multiplexed high-throughput fluorescence microscopy of fixed cells followed by feature extraction and morphological profiling to search for features that contribute to a signature of drug resistance.”

[Text revision] We clarified that the Bortezomib Signature did not correspond to well position (originally lines #155-157).

Reviewer 1 - Minor Comment 9Line #155-156: "We found that the pattern of Bortezomib Signatures corresponded to the cell identity plate layout", the word "not" is missing before "corresponded".

“We found that the pattern of Bortezomib Signatures did not correspond to well position relative to the plate (Fig. 2-Supplement 7 B), indicating that the well position for each clone was not strongly contributing to its Bortezomib Signature.”

[Text revision] We explicitly described the result that some misclassified clones (WT10, WT15, and BZ06) did not have unexpected bortezomib sensitivity as determined by proliferation assays. We also moved the supplementary figure to an updated Figure 3 to better highlight this result (described below in “Figure revisions already carried out”). Lastly, we add a new figure (Figure 5-Supplement 1) to more explicitly analyze the misclassified lines (described below in “New analyses already carried out”).

Reviewer 3 - Minor Comment 3The bortezomib sensitivity of the WT lines used in the last experiments was determined and did not seem to be greater than parental. This could be mentioned in the text; the figure raises the question and the answer is provided, but it's in the supplemental material.

“While the Bortezomib Signature correctly characterized the bortezomib sensitivity of most clones, it consistently misclassified others (WT10, WT15, and BZ06) (Fig 5-Supplement 1 A). Proliferation assays conducted in earlier experiments showed that WT10 and WT15 were sensitive to bortezomib while BZ06 was resistant (Fig. 1-Supplement 2 A and B). By comparing these incorrect predictions with high-confidence correct predictions, we observed differences that varied by clone type, suggesting unique morphology may be driving each of these misclassifications (Fig. 5-Supplement 1 B and C). These results are consistent with the Bortezomib Signature being generalizable to clones not included in the training dataset and suggest that morphological profiling has the potential to identify bortezomib-resistant clones based on the morphological features of cells in the absence of drug treatment.”

[Text revision] We clarified that the metrics (accuracy and average precision) were based on median Bortezomib Signature scores of all replicate well-level profiles per clone. We can compare samples based on rank, and difference from 95% confidence interval of permuted data. There is no current way for our method to assign a likelihood. Also note that we have updated the discussion to discuss alternative metrics (see Reviewer 1 - Minor Comment 7) These are very important distinctions, and we are grateful to the reviewer for bringing them up.

Reviewer 3 - Major Comment 3The study classifies cells as binary sensitive or resistant, but would results be improved by scoring based on likelihood of being resistant/sensitive?Reviewer 3 - Minor Comment 2It is not clear whether the accuracy was based on a percentage of replicates per cell line that were classified correctly or whether that was referring to classification of the cell line overall as sensitive/resistant.

“We next examined whether the Bortezomib Signature was able to predict the bortezomib resistance of a clone based on morphological profiling data (Fig. 3 A-E and Fig. 3-Supplement 2 A and B). We called the clone bortezomib-resistant if the median Bortezomib Signature of all replicate well profiles was greater than zero and bortezomib-sensitive if the median Bortezomib Signature less than zero. In the training dataset, the Bortezomib Signature correctly predicted the bortezomib resistance of all ten clones, with median Bortezomib Signatures for eight out of ten clones beyond the 95% confidence interval for the randomly permuted data (Fig. 3 A). The accuracy of the Bortezomib Signature was 88% while the average precision was 81% for the training dataset (Fig. 3-Supplement 2 A and B) (see Methods). The signature performed similarly well in the validation dataset (Fig. 3 B), with an accuracy of 92% and an average precision of 89% (Fig. 3-Supplement 2 A and B). In the test dataset the Bortezomib Signature correctly predicted the bortezomib resistance of all clones, though only HCT116 parental cells had a median Bortezomib Signature outside the 95% confidence interval for the randomly permuted data (Fig. 3 C). The test dataset had an accuracy of 80% and an average precision of 68% (Fig. 3-Supplement 2 A and B). Similarly, in the holdout dataset the Bortezomib Signature had an accuracy of 78% and an average precision of 69% (Fig.3 -Supplement 2 A and B), and correctly predicted the bortezomib resistance of twelve out of thirteen clones, with WT01 misclassified as bortezomib-resistant (Fig. 3 D). In the holdout dataset, four of the twelve correctly characterized clones had median Bortezomib Signatures outside the 95% confidence interval for the randomly permuted data.”

We also mirrored language when discussing the ixazomib and CB-5083 results.

“However, only two of the four correctly identified ixazomib-resistant clones and one of the three CB-5083-resistant clones had median Bortezomib Signatures outside the 95% confidence interval of the randomly permuted data. The area under the ROC (AUROC) curve for ixazomib-resistant and CB-5083-resistant clones (0.63 and 0.60, respectively) was lower than those calculated for the training, validation, test, and holdout datasets. In addition, many of the Bortezomib Signatures for well profiles of ixazomib- and CB-5083-resistant clones, particularly those for CB-5083-resistant clones, landed within the 95% confidence interval of the randomly permuted data. These results suggest that the Bortezomib Signature is not a general signature of UPS-targeting drug resistance and instead has some specificity for bortezomib.”

[Text revision] We added an explicit note that our image analysis pipelines are also publicly available. Our reporting of our data processing pipelines are documented fully and well above standards in our field. Linking the publicly-available resources with these methods maximizes reproducibility.

Reviewer 1 - Minor Comment 10Additional details on the processing steps in the analysis pipeline in the Methods will be highly appreciated.

“We include all image analysis pipelines at https://github.com/broadinstitute/profiling-resistance-mechanisms (G. Way et al. 2023).”

[Text revision] We have compared our approach to the on-disease/off-disease scores as introduced in (Heiser et al. 2020). We agree with the reviewer that a discussion of these two methods would help clarify our phenotypic signature concept. The on/off score is about the degree to which a perturbation pushes disease towards a healthy state. In this case we have 3 sets of data: healthy samples (used for training), disease samples (used for training), and the sample we want to score, which should be of the form "disease + perturbation". With our approach, based on singscore, we also have 3 sets of data: sensitive samples (used for training), resistance samples (used for training), and the sample we want to score. Here, our sample we want to score could be anything, not necessarily of the form "resistance + perturbation". Furthermore, singscore does not have the concept of orthogonality to resistance/sensitivity. This would become relevant if we were exploring perturbations or conditions that would induce a resistant cell line to become sensitive, but we are not doing that here. There are other statistical differences (projection vs. rank based etc.) but the key difference is the applicability of the method to the specific problem at hand.

Reviewer 1 - Minor Comment 7How is the Bortezomib Signature related to the "on-disease"/"off-disease" scores described in https://www.biorxiv.org/content/10.1101/2020.04.21.054387v1.full? Are there other alternatives used for similar binary phenotypic signatures? What is the justification for using these measurements? I would love to see this generalized concept explicitly discussed in the Discussion.

We added the following to the discussion.

“The Bortezomib Signature is conceptually similar to the on-disease/off-disease score (Heiser et al. 2020). Both require three phenotypic measurements: a target phenotype representing ideal, a disease phenotype, and a new phenotype to classify. However, our approach is technically different (non-parametric compared to linear projection) and our goals are different (phenotypic classification compared to perturbation alignment). Other methods also enable phenotype labeling, but they focus on single-sample annotation without regard to a target phenotype (Wawer et al. 2014; Rohban et al. 2017; Simm et al. 2018; Nyffeler et al. 2020).”

Figure revisions already carried out

[Figure revision] We moved all boxplots from the original Fig. 2-Supplement 9 to the main text (also splitting Fig. 2 into Fig. 2 and 3). From the original Figure 2, we moved the accuracy and average precision bar graphs to the supplement. We also note that this change increases transparency of the discriminative signal of our signature.

Reviewer 1 - Minor Comment 8I would highly recommend showing the Bortezomib Signatures from Figure 2-Supplement 9. in Fig. 2. This was the main measurement used throughout the manuscript and in my opinion, it is very important to consistently visualize the data along the manuscript, for clarity and easier reader interpretation.

[Figure revision] We adjusted the position of the legend in the accuracy and average precision bar graphs (originally Fig. 2 C and D, now Fig. 3-Supplement 2) for clarity. We also note that keeping the bar chart here is standard best practice (compared to a dot plot).

Reviewer 1 - Minor Comment 4I found the visualization in Fig. 2C-D not intuitive (it is properly explained in the legend). I suggest replacing the accuracy colorbar with a color marker to make it more distinct from the random permutation (|--*--|) The location of the text "mean +- SD of 100 random permutation" made me first think that it is linked to the holdout.

[Figure revision] We changed the point distribution in the boxplots (from expanded to standard) to minimize overlap with the boxplot lines. We also updated the legend text to indicate that individual points in boxplots represent the Bortezomib Signature for well profiles. Note, we paste a representative example of this change above (new Figure 3).

Reviewer 3 - Minor Comment 1I found the box plots somewhat difficult to interpret (especially where the WT lines had a lot of overlap with the red shaded area). Do the points in these charts correspond to replicate wells?

We also update the figure legend.

“Plots show values for individual well profiles (points), range (error bars), 25th and 75th percentiles (box boundaries), and median.”

[Figure revision] [Response to Reviewer 2 - Major Comment 7] We thank the reviewer for allowing us an opportunity to clarify the mechanism. We feel that it is beyond scope of this manuscript to disentangle the molecular alterations that cause bortezomib resistance based on our Cell Painting insights. This wet lab experimental process is arduous and cost prohibitive, and we argue that one of the benefits of taking a morphology approach to resistance status is that we can detect resistant cells (and therefore cells that won’t die when presented with a treatment) without knowing the molecular mechanism.

Nevertheless, the reviewer has encouraged us to enhance the ability for a reader to view and interpret the signature to perhaps more easily facilitate future work. Previously, we presented our signature in text form in Figure 2-Supplement 4 and in heatmap form in Figure 2-Supplement 5. Here, we add a new figure (Figure 2-Supplement 6; pasted below) which will improve interpretability.

Reviewer 2 - Major Comment 7:Next to feature importance, the authors do not discuss (or I missed) what biology the features represent. Such the reader is left wondering what the actual mechanism of bortezomib resistance could be and if cell painting could shed light on the molecular alterations that cause the treatment resistance. While reviewing, I thus wondered which audience the authors targeted with their manuscript. A more focused analysis of their data that highlights aspects of the study either for the machine learning community, the cell biology community, or the precision oncology community would greatly benefit the manuscript's impact. In its current form, the study's findings seem diluted and spread across a wide range of research questions.

Additionally, we add the following to the results section:

“We then examined the grouping of features across compartments and channels and found radial distribution features were higher in resistant cells (Fig 2-Supplement 6).”

The code change to generate the signature visualization summary is available at: https://github.com/broadinstitute/profiling-resistance-mechanisms/pull/131

*New analyses already carried out*

[New analysis] [Response to Reviewer 2 - Major Comment 5] We agree that a systematic analysis of feature selection methods will provide additional insights not already in the manuscript. Therefore, we have performed two new computational experiments to compare our linear modeling feature selection approach against other standard approaches. We demonstrate that our linear modeling approach is effective at isolating the core differences between resistant and sensitive classes.

Specifically, we performed two analyses: (A) UMAP and (B) k-means cluster analysis. We analyzed profiles defined by four different feature selection approaches: (1) Using all traditional CellProfiler features; (2) Using the traditional CellProfiler feature selection approach (removing low variance features, high correlating features, etc.); (3) Using 45 random features (same size as Bortezomib Signature); and (4) Using only the bortezomib signature features. We performed Fisher’s exact tests to derive odds ratios of cluster membership by resistance status and calculated Silhouette widths to quantify relative proximity of clusters.

This analysis generates a new supplementary figure (see below), and demonstrates that the linear-modeling-based feature selection isolated the features driving the differences between the clone types (resistance vs. wildtype) while the standard approaches do not as effectively separate.

Reviewer 2 - Major Comment 5:A fascinating bit of the manuscript is the description of the feature selection from the screen is done systematically, considering the technical and biological variability and technical artifacts and modeling covariates using linear models seems a very appropriate way of doing so and could serve as another proof of concept that this is indeed the most robust way of modeling and removing signal of technical covariates from the data. Yet, I wondered why the authors do not discuss other means of feature selection or dimensionality reduction; further, they need to show how the features cluster the cell lines or why impact (information content) different features deliver. For an audience interested in the technical aspects of cell painting analysis and machine learning based on the data, that would, IMHO, be the most exciting questions.

Additionally, we add the following to the results section:

“We then compared our linear-modeling approach to feature selection against other feature spaces and found that the Bortezomib Signature clusters same-type clones (bortezomib-resistant vs. bortezomib-sensitive) with higher enrichment compared to the full feature space, standard feature selection (see Methods), or a random selection of 45 features (Fig 3-Supplement 3).”

And methods section, describing this analysis:

“We were also interested in comparing the ability of different feature spaces to cluster clones of the same type (resistant vs. sensitive). This analysis would determine if the Bortezomib Signature features, which we derived using linear modeling to isolate biological from technical variables, had a greater ability to cluster. We compared the Bortezomib Signature against three other feature spaces: (1) the full feature space, (2) standard feature selection (see Image data processing methods), and (3) 45 randomly selected features. We performed two analyses using these four feature spaces including Uniform Manifold Approximation and Projection (UMAP) (McInnes et al. 2018) and k-means clustering. For UMAP, we used default umap-learn parameters to identify two UMAP coordinates per feature space. We then visualized the clusters by their resistance status and Bortezomib Signature score. The UMAP analysis represents a qualitative analysis. Next, we applied k-means clustering with 25 initializations across a range of 2-14 clusters (k). Prior to clustering and for each feature space, we applied principal component analysis (PCA) and transformed each feature space into 30 principal components. This step was necessary to compare k-means clustering metrics, which are sensitive to the feature space dimensionality. We applied a Fisher’s exact test to each cluster using a two-by-two contingency matrix that specified cluster membership for each clone classification (resistant vs. sensitive). We visualized the mean odds ratio and max cluster odds ratio for each feature space across k. A high odds ratio tells us that the feature space effectively clusters clones of the same resistance status. Lastly, we calculated Silhouette width (the average proximity between samples in one cluster to the second nearest cluster) for each feature space across k.”

The code change to derive the UMAP coordinates, perform clustering, and generate the figure is available at https://github.com/broadinstitute/profiling-resistance-mechanisms/pull/132

[New analysis] [Response to Reviewer 3 - Major Comment 1] We thank the reviewer for this suggestion, which allowed us to explore the misclassified samples in more depth. We added a new supplementary figure in which we summarized all bortezomib clones (wildtype and resistant) in their accuracy based on the bortezomib signature (panel A). We did not include training set samples in this analysis. Using samples that were consistently incorrectly classified with high confidence (three samples: WT15, BZ06, WT10) we performed two separate two-sample Kolmogorov–Smirnov (KS) tests. Specifically, we compared high incorrect wildtype to high correct wildtype and high incorrect resistant to high correct resistant. Our results indicate that most bortezomib signatures were significantly different between correct and incorrect assignments (panel B), and that the signature features varied between resistant and wildtype misclassification tests (panel C).

Reviewer 3 - Major Comment 1:While the claims are largely substantiated, there are a few points where further consideration would improve the manuscript. Several cell lines were mis-classified with what appears to be a high degree of certainty. Can the authors tell what was driving those predictions? Was there something in the morphological signature that weighed more heavily in those cases?

Additionally, we add the following to the results section:

“While the Bortezomib Signature correctly characterized the bortezomib sensitivity of most clones, it consistently misclassified others (WT10, WT15, and BZ06) (Fig 5-Supplement 1 A). Proliferation assays conducted in earlier experiments showed that WT10 and WT15 were sensitive to bortezomib while BZ06 was resistant (Fig. 1-Supplement 2 A and B). By comparing these incorrect predictions with high-confidence correct predictions, we observed differences that varied by clone type, suggesting unique morphology may be driving each of these misclassifications (Fig. 5-Supplement 1 B and C). These results are consistent with the Bortezomib Signature being generalizable to clones not included in the training dataset and suggest that morphological profiling has the potential to identify bortezomib-resistant clones based on the morphological features of cells in the absence of drug treatment.”

And methods section, describing this analysis:

“Some profiles were consistently predicted incorrectly with high confidence but in the opposite direction (see Figure 5-Supplement 1). For a well-level profile to be categorized as high-confidence (in either the correct or incorrect directions), it needed to score beyond the 95% confidence interval of the randomly permuted data range. For example, a high-confidence incorrect resistant profile would have a Bortezomib Signature below 95% confidence interval of the randomly permuted data. To evaluate the features driving the differences in these samples, we applied two-sample Kolmogorov–Smirnov (KS) tests per Bortezomib Signature feature. We applied these tests to two separate groups: (1) misclassified bortezomib-sensitive vs. high-confidence accurate bortezomib-sensitive and (2) misclassified bortezomib-resistant vs. high-confidence accurate bortezomib-resistant.”

The code change to generate the UMAP coordinates and figure is available at https://github.com/broadinstitute/profiling-resistance-mechanisms/pull/130

[Response to Reviewer 2 - Minor Comments 1 and 2]: These are interesting suggestions! Still, we prefer not to speculate on the biological mechanism of the Bortezomib signature.

Connecting morphological features identified as contributing to the Bortezomib Signature by Cell Painting to specific biological pathways would demand considerable cell-based assays to validate. In addition, our analyses suggest that the features contributing to the Bortezomib Signature are spread across a range of cellular compartments and channels, making it difficult to pin down specific mechanisms or pathways as likely contributors to bortezomib resistance. However, we are adding a figure to increase interpretability of the signature, which will aid in developing future hypotheses. Note that the signature was not possible to detect by eye (Fig. 2 A).

Reviewer 2 - Minor Comment 1:There could be some speculation on the mechanism of Bortezomib resistance concerning the literature with the existing image data. For example, Bortezomib resistance is connected to serine synthesis and how a particular feature could contribute to the known mechanism.Reviewer 2 - Minor Comment 2:Along the same lines, the authors could show that larger cells lead to resistance with microscopic images.

[Response to Reviewer 2 - Major Comment 8]: We appreciate the reviewer’s concern that our work using HCT116 clonal cells lines may not directly reflect results from patient samples. Our choice was based on previously published work demonstrating the efficiency with which HCT116 cells generate resistant clones due to diminished DNA mismatch repair and decreased expression of drug efflux pumps. Since our work is a proof of concept rather than a comprehensive demonstration of translating morphological profiling into clinical practice, we believe that experiments using multiple patient cell lines from different tissues as well as digital pathology records to be beyond the scope of this work. We instead chose to tone down the language of our manuscript to more clearly acknowledge the limitations of our work and clarify this as a proof of concept.

Reviewer 2 - Major Comment 8 (relevant excerpt):I suggest the authors test their approach on at least two other cell lines (maybe from different tissues) and benchmark their results against a dataset of digital pathology where such predictions are made from stained and analyzed tissue slices. This way, after a thorough benchmark against related third-party data sets, the method would significantly gain relevance, the paper would appeal to a broader audience, and the advance gains more merit.

[Response to Reviewer 3 - Major Comment 2]: The bortezomib sensitivity of ixazomib- and CB-5083-resistant clones was not determined, and hence can not be ruled out as a possible explanation for their high Bortezomib Signature scores. However, we prefer not to conduct additional proliferation assays for the misclassified clones (IX02, WT06, CB14, CB16) in the presence of bortezomib to determine whether coincidental bortezomib resistance might explain the signature performance. Our rationale is that three other misclassified clones (WT10, WT15, and BZ06) had the expected bortezomib sensitivity in proliferation assays (Fig. 1-Supplement 2), meaning that additional proliferation assays may not reveal any insights regarding the signature performance.

Reviewer 3 - Major Comment 2:Was the bortezomib sensitivity of the IX (or CB) resistant cell lines determined? If there were differences, this could explain some of the variation in the morphological signatures. This could be easily done in one or two growth experiments.

[Response to Reviewer 2 - Major Comment 7]: Thank you for pointing this out. Our goal is to keep the study multi-disciplinary. We are adding a figure to increase interpretability of the signature, and adding text-based clarifications.

Reviewer 2 - Major Comment 7 (relevant excerpt):While reviewing, I thus wondered which audience the authors targeted with their manuscript. A more focused analysis of their data that highlights aspects of the study either for the machine learning community, the cell biology community, or the precision oncology community would greatly benefit the manuscript's impact. In its current form, the study's findings seem diluted and spread across a wide range of research questions.

[Response to Reviewer 2 and 3 - Major Comments 6 and 4]: We prefer not to expand the scope of the model to predict other drug signatures. This would require a substantial amount of work to generate the appropriate drug-resistant clones, collect the imaging data, and analyze it, and we think it important to convey the purpose of our paper is proof of concept. We do not feel that the time invested in performing this analysis would result in adequate returns beyond what we already demonstrate.

Reviewer 2 - Major Comment 6.Interestingly, the Bortezomib signature is specific to the drug and not a broad range of proteasomal inhibitors. However, seeing the common features between all the proteasomal inhibitors would be interesting.

Reviewer 3 - Major Comment 4There was some predictive ability of the Bortezomib Signature for ixazomib resistance. Were there some features that were correlated with IX-resistance, i.e. UPS pathway, versus specific to bortezomib? Do the features suggest anything about resistance mechanisms or is the feature set too abstruse to interpret?

References

Foroutan, Momeneh, Dharmesh D. Bhuva, Ruqian Lyu, Kristy Horan, Joseph Cursons, and Melissa J. Davis. 2018. “Single Sample Scoring of Molecular Phenotypes.” *BMC Bioinformatics* 19 (1): 404.

Heiser, Katie, Peter F. McLean, Chadwick T. Davis, Ben Fogelson, Hannah B. Gordon, Pamela Jacobson, Brett Hurst, et al. 2020. “Identification of Potential Treatments for COVID-19 through Artificial Intelligence-Enabled Phenomic Analysis of Human Cells Infected with SARS-CoV-2.” *bioRxiv*. https://doi.org/10.1101/2020.04.21.054387.

McInnes, Leland, John Healy, Nathaniel Saul, and Lukas Großberger. 2018. “UMAP: Uniform Manifold Approximation and Projection.” *Journal of Open Source Software* 3 (29): 861.

Nyffeler, Johanna, Clinton Willis, Ryan Lougee, Ann Richard, Katie Paul-Friedman, and Joshua A. Harrill. 2020. “Bioactivity Screening of Environmental Chemicals Using Imaging-Based High-Throughput Phenotypic Profiling.” *Toxicology and Applied Pharmacology* 389 (January): 114876.

Rohban, Mohammad Hossein, Shantanu Singh, Xiaoyun Wu, Julia B. Berthet, Mark-Anthony Bray, Yashaswi Shrestha, Xaralabos Varelas, Jesse S. Boehm, and Anne E. Carpenter. 2017. “Systematic Morphological Profiling of Human Gene and Allele Function via Cell Painting.” *eLife* 6 (March). https://doi.org/10.7554/eLife.24060.

Simm, Jaak, Günter Klambauer, Adam Arany, Marvin Steijaert, Jörg Kurt Wegner, Emmanuel Gustin, Vladimir Chupakhin, et al. 2018. “Repurposing High-Throughput Image Assays Enables Biological Activity Prediction for Drug Discovery.” *Cell Chemical Biology* 25 (5): 611–18.e3.

Wacker, Sarah A., Benjamin R. Houghtaling, Olivier Elemento, and Tarun M. Kapoor. 2012. “Using Transcriptome Sequencing to Identify Mechanisms of Drug Action and Resistance.” *Nature Chemical Biology* 8 (3): 235–37.

Wawer, Mathias J., Kejie Li, Sigrun M. Gustafsdottir, Vebjorn Ljosa, Nicole E. Bodycombe, Melissa A. Marton, Katherine L. Sokolnicki, et al. 2014. “Toward Performance-Diverse Small-Molecule Libraries for Cell-Based Phenotypic Screening Using Multiplexed High-Dimensional Profiling.” *Proceedings of the National Academy of Sciences of the United States of America* 111 (30): 10911–16.

Way, Gregory, Yu Han, David Stirling, and Shantanu Singh. 2023. Broadinstitute/profiling-Resistance-Mechanisms: Analysis for Preprint. Zenodo. https://doi.org/10.5281/ZENODO.7803787.

Way, Gregory P., Maria Kost-Alimova, Tsukasa Shibue, William F. Harrington, Stanley Gill, Federica Piccioni, Tim Becker, et al. 2021. “Predicting Cell Health Phenotypes Using Image-Based Morphology Profiling.” *Molecular Biology of the Cell* 32 (9): 995–1005.